# The conserved ATPase PCH-2 controls the number and distribution of crossovers by antagonizing their formation in *Caenorhabditis elegans*

**Bhumil Patel, Maryke Grobler, Alberto Herrera, Elias Logari, Valery Ortiz, Needhi Bhalla\***

Department of Molecular, Cell and Developmental Biology, University of California, Santa Cruz, Santa Cruz, United States

## eLife Assessment

This is an **important** study examining the role of conserved PCH-2 protein at different stages of *C. elegans* meiosis. The authors use elegant molecular genetic approaches to provide **convincing** evidence to support their claims. The work will be of interest to scientists studying meiosis, DNA recombination, and chromosome segregation.

**\*For correspondence:**
nbhalla@ucsc.edu

**Competing interest:** The authors declare that no competing interests exist.

## Abstract

Meiotic crossover recombination is essential for both accurate chromosome segregation and the generation of new haplotypes for natural selection to act upon. This requirement is known as crossover assurance and is one example of crossover control. While the conserved role of the ATPase, PCH-2, during meiotic prophase has been enigmatic, a universal phenotype when *pch-2* or its orthologs are mutated is a change in the number and distribution of meiotic crossovers. Here, we show that PCH-2 controls the number and distribution of crossovers by antagonizing their formation. This antagonism produces different effects at different stages of meiotic prophase: early in meiotic prophase, PCH-2 prevents double-strand breaks from becoming crossover-eligible intermediates, limiting crossover formation at sites of initial double-strand break formation and homolog interactions. Later in meiotic prophase, PCH-2 winnows the number of crossover-eligible intermediates, contributing to the designation of crossovers and ultimately, crossover assurance. We also demonstrate that PCH-2 accomplishes this regulation through the meiotic HORMAD, HIM-3. Our data strongly support a model in which PCH-2's conserved role is to remodel meiotic HORMADs throughout meiotic prophase to destabilize crossover-eligible precursors and coordinate meiotic recombination with synapsis, ensuring the progressive implementation of meiotic recombination and explaining its function in the pachytene checkpoint and crossover control.

## Introduction

Meiosis is a specialized type of cell division that reduces chromosome number by half, resulting in the production of genetically diverse haploid gametes, so that fertilization during sexual reproduction restores diploidy. This process occurs in two stages: meiosis I, in which homologous chromosomes are partitioned, and meiosis II, in which sister chromatid are segregated. The regulation of meiosis is crucial for ensuring that both genetic recombination and chromosome segregation occur accurately. Errors in human meiosis are associated with birth defects, such as Down and Turner syndromes,

infertility and miscarriages, underscoring the importance of understanding meiosis to human health (*Hassold and Hunt, 2001*).

During prophase I, homologous chromosomes pair and this pairing is stabilized through a process called synapsis, in which a protein structure called the synaptonemal complex (SC) holds homologs together. This close association between homologs during synapsis facilitates crossover formation, the process in which double-strand breaks (DSBs) are deliberately introduced into the genome, repaired by meiosis-specific mechanisms that exchange DNA between homologous chromosomes and generate the chiasmata, or linkage, that promotes accurate meiotic chromosome segregation. Therefore, disruptions in pairing, synapsis, or recombination prevents the formation of chiasmata and can lead to meiotic errors such as nondisjunction, resulting in aneuploid gametes.

In addition to the fundamental role that recombination plays in ensuring accurate chromosome segregation, crossover recombination accomplishes another important function: it generates new haplotypes for natural selection to act upon to drive evolution. Thus, to assure a random assortment of alleles on a population level, the distribution of crossovers may be as tightly regulated as their number (*Veller et al., 2019*). The significance of controlling both crossover number and distribution is clearly illustrated by the existence of mechanisms such as crossover assurance, in which every pair of homologous chromosomes gets at least one crossover; crossover homeostasis, in which the number of crossovers remains relatively invariant even if the number of recombination precursors change; and crossover interference, in which the presence of a crossover inhibits the formation of a crossover nearby (*Gray and Cohen, 2016*). DSBs typically vastly outnumber crossovers in most organisms and are introduced gradually throughout early prophase (*Joshi et al., 2015*; *Woglar and Villeneuve, 2018*). Therefore, to accomplish this precise level of control, meiotic crossover recombination and the decision about which DSBs become crossover-eligible intermediates, and eventually, which crossover-eligible intermediates get designated as crossovers, is implemented progressively throughout meiotic prophase (*Cole et al., 2012*; *Joshi et al., 2015*; *Morgan et al., 2021*; *Yokoo et al., 2012*). In many systems, transitions from DSBs to crossover-eligible intermediates, and crossover-eligible intermediates to crossovers, can be molecularly and/or cytologically monitored.

PCH-2, also known as TRIP13 in mammals, is an evolutionarily ancient AAA-ATPase that plays a significant role in regulating meiosis across different organisms, including *Mus musculus* (mice), *Saccharomyces cerevisiae* (budding yeast), *Drosophila melanogaster* (fruit flies), and *Caenorhabditis elegans* (worms) (*Bhalla, 2023*). PCH-2 and its orthologs structurally remodel a family of proteins with HORMA domains (HORMADs) to control their function (*Gu et al., 2022*). HORMADs participate in a variety of signaling events and can exist in at least three structurally distinct conformations: a 'closed' conformation, which they adopt when they bind a short peptide sequence in their own protein sequence or another protein (also called a closure motif) and their C-terminus wraps around this motif to stabilize the interaction; an 'open' conformation when unbound, in which their C-terminus is discretely tucked against the HORMA domain; and an 'extended' conformation, which is an intermediate between the two (*Gu et al., 2022*). PCH-2 and its orthologs convert the closed version of HORMADs to the open or extended versions, playing an important role in recycling HORMADs during signaling. During meiosis, closed versions of meiotic HORMADs assemble on chromosomes to form meiotic chromosome axes, which are essential for pairing, synapsis, and recombination between homologous chromosomes (*Kim et al., 2014*). The remodeling of meiotic HORMADs by PCH-2 to an 'open' or 'extended' conformation is thought to reduce the levels of HORMADs on chromosomes (*Börner et al., 2008*; *Cuacos et al., 2021*; *Lambing et al., 2015*; *Wojtasz et al., 2009*) and/or increase their dynamic association and dissociation (*Russo et al., 2023*), modulating and coordinating homolog pairing, synapsis and recombination during prophase.

Despite being discovered as a pachytene checkpoint component in 1999 (*San-Segundo and Roeder, 1999*), the conserved meiotic role of the PCH-2/HORMAD module has been difficult to characterize, in part, we have argued, because of the evolutionary innovation that is an inherent aspect of sexual reproduction (*Bhalla, 2023*). Moreover, in some systems, such as budding yeast and plants (*Herruzo et al., 2021*; *Yang et al., 2020*), PCH-2 orthologs not only remodel meiotic HORMADs on meiotic chromosomes but also perform this function in the cytoplasm to make meiotic HORMADs available for their role(s) in meiotic nuclei. This dual role can complicate functional analyses, particularly in *pch-2* null mutants. However, all meiotic systems exhibit defects in the number and distribution of crossovers when PCH-2 function is abrogated by mutation (*Deshong et al., 2014*; *Joshi et al.,*

*2009*; *Joyce and McKim, 2009*; *Lambing et al., 2015*; *Roig et al., 2010*; *Zanders and Alani, 2009*). Unfortunately, the lack of a clear pattern when analyzing these defects in recombination in *pch-2* mutants has contributed to an inability to develop a unified, integrated model of PCH-2 function in the field.

In *C. elegans*, meiotic HORMADs localize to meiotic chromosomes independently of PCH-2 (*Deshong et al., 2014*). Moreover, the progressive implementation of meiotic recombination can be cytologically monitored in meiotic nuclei that are organized both spatially and temporally in the *C. elegans* germline (*Woglar and Villeneuve, 2018*; *Yokoo et al., 2012*). Here, we exploit this system to show that PCH-2 is required to control the number and distribution of crossovers by antagonizing their formation. This antagonism produces different consequences depending on the stage of meiotic prophase. In early meiotic prophase, PCH-2 inhibits DSBs from becoming crossover-eligible intermediates, ensuring that crossovers are more widely distributed than sites of initial DSB formation and/or homolog interactions. Later in meiotic prophase, PCH-2 is responsible for winnowing the numbers of crossover-eligible intermediates on synapsed chromosomes, contributing to the designation of crossovers and ultimately, crossover assurance. Genetic analysis demonstrates that PCH-2's regulation of crossover-eligible intermediates is through one of three essential meiotic HORMADs, HIM-3. Finally, we link PCH-2's effect on early DSBs in early meiotic prophase to cell cycle stage, demonstrating that both limit early DSBs from becoming crossovers. We propose that PCH-2's remodeling of HIM-3 on meiotic chromosomes destabilizes crossover-eligible intermediates throughout meiotic prophase, contributing to the progressive implementation of meiotic recombination to control the number and distribution of crossovers, also known as crossover control.

## Results

### PCH-2 controls the number and distribution of crossovers in similar patterns on multiple chromosomes

We had previously shown that loss of PCH-2 reduced the frequency of double crossovers and genetic length of both an autosome (chromosome III) and the X chromosome, albeit not uniformly among genetic intervals (*Deshong et al., 2014*). To determine whether a more obvious pattern could be observed, we expanded our analysis by monitoring recombination genetically in both wildtype and *pch-2* mutant animals using five single-nucleotide polymorphisms (SNPs) that spanned 95% of chromosomes, I, III, IV, and the X chromosome (*Figure 1*). We excluded chromosome II from our analysis because of the potential difficulty combining Hawaiian SNPs with the *pch-2* mutation, which is linked to chromosome II, and chromosome V because of our use of the *bcIs39* transgene to identify cross progeny, which may disrupt recombination on that chromosome.

In wildtype animals, we observed multiple double crossovers, ranging from 1 to 13, depending on the chromosome. In a majority of these double crossovers (69%), one crossover was at the end of the chromosome where pairing and synapsis initiate, also called the pairing center (PC) (*MacQueen et al., 2005*), suggesting some relationship between where chromosomes might make initial contacts and the likelihood of double crossovers. On all chromosomes analyzed, we observed no double crossovers in *pch-2* mutants and this difference was statistically significant for chromosomes I, III, and X. Moreover, there was a striking and consistent shift of crossovers to the PC end of all four chromosomes tested. This shift in the distribution of crossovers to the PC ends of chromosomes was generally accompanied by a reduction in crossovers in the center of chromosomes. In *C. elegans*, the center of chromosomes are where DSBs are less numerous (*Nadarajan et al., 2021*; *Yu et al., 2016*) and where genes are more abundant (*Consortium, 1998*). In the case of the X chromosome and chromosome III, recombination at the non-PC end was also reduced and appeared to more closely resemble the physical map at this end of these chromosomes. Thus, PCH-2 ensures a wider distribution of crossovers across chromosomes, away from regions that are more likely to undergo early homolog interactions (PC ends) and with more DSBs (both the PC and non-PC ends of chromosomes), and toward the center of chromosomes, where DSBs are less abundant.

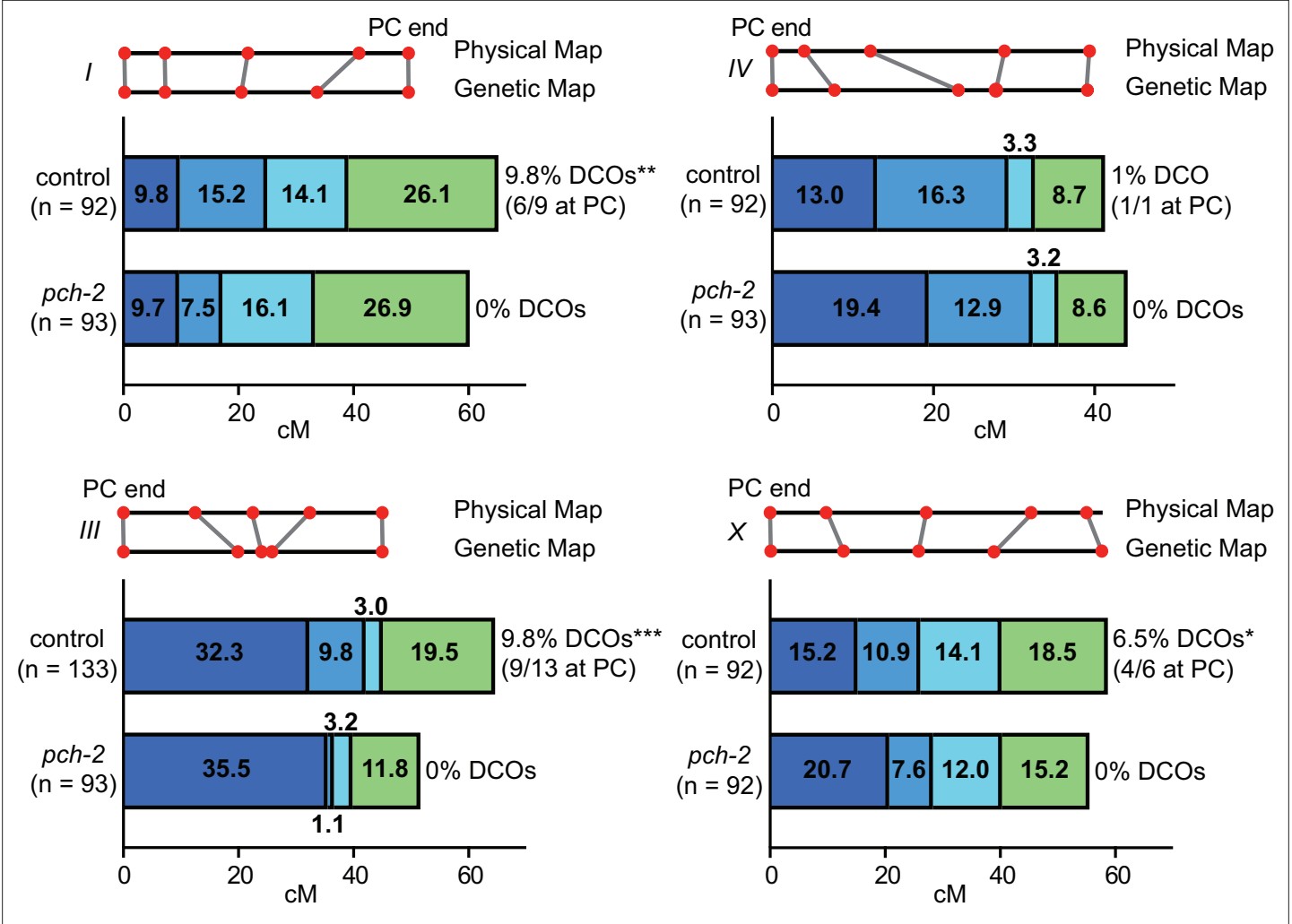

**Figure 1.** PCH-2 controls the number and distribution of crossovers in similar patterns on multiple chromosomes. Genetic analysis of meiotic recombination in wildtype and *pch-2* mutants. DCO indicates double crossovers. Physical and genetic maps of chromosomes I, III, IV and the X chromosome are depicted to scale. Genetic distance is shown in centimorgans. *p<0.05, **p<0.01, and ***p<0.001.

The online version of this article includes the following source data for figure 1:

**Source data 1.** Numerical data depicted in *Figure 1*.

## PCH-2 prevents exogenous DSBs early in meiotic prophase from becoming crossovers

We previously showed that PCH-2 promotes crossover formation and crossover assurance through its regulation of the meiotic HORMAD, HIM-3, in *C. elegans* (*Russo et al., 2023*). Moreover, this promoting role seems linked to its localization to the SC (*Patel et al., 2023*). However, the observed shift in the distribution of crossovers in *pch-2* mutants suggests that PCH-2 may also play a role in inhibiting crossovers and that this role may be occurring in early meiotic prophase, when chromosomes are undergoing initial homolog interactions. We had previously observed that meiotic nuclei in early prophase were more likely to produce crossovers when DSBs were induced by excision of the *Mos* transposon in *pch-2* mutants than in control animals but experimental caveats limited our ability to properly interpret this experiment (*Deshong et al., 2014*).

To explicitly test this possibility, we took advantage of both the spatiotemporal organization of meiotic nuclei in the *C. elegans* germline and the observation that nuclei travel in an assembly line process at a stereotypical pace to late pachytene (*Jaramillo-Lambert et al., 2007*), where we can cytologically assess crossover formation by staining for the essential crossover factor, COSA-1 (*Yokoo*

*et al., 2012*). We performed irradiation experiments to introduce exogenous DSBs at different time-points during meiotic prophase and analyzed PCH-2's role in crossover formation (*Figure 2A*).

In this experiment, we tested the effects of ionizing radiation at two distinct timepoints: 8 hours after irradiation, when we can monitor crossover formation in late pachytene nuclei that received exogenous DSBs in mid-pachytene, and 24 hours after irradiation, when we monitor crossover formation in late pachytene nuclei that received exogenous DSBs in leptotene/zygotene, also known as the transition zone in *C. elegans* (*Figure 2A*). In order to detect crossovers in late pachytene, control and *pch-2* mutants germlines were stained for GFP::COSA-1, which allowed us to visualize crossovers, and DAPI, which allowed us to visualize DNA. In unirradiated control worms, 94% of late pachytene nuclei have six GFP::COSA-1 foci, one per chromosome pair (*Figure 2B*). In unirradiated *pch-2* mutants, the percentage of nuclei have six GFP::COSA-1 foci drops to 82%, primarily because of the significant increase in meiotic nuclei with less than six GFP::COSA-1 foci (17%) (*Figure 2B*), consistent with our previous report that loss of PCH-2 leads to a decrease in crossover formation (*Deshong et al., 2014*). At 8 hours after irradiation, *pch-2* mutants' defect in crossover assurance becomes even more pronounced. We observed a significant increase in nuclei containing less than six COSA-1 nuclei in *pch-2* mutants (26%) compared to control (9%) germlines (*Figure 2C and D*). However, we did not observe a significant number of nuclei with greater than six COSA-1 foci in either backgrounds. This observation reinforces previous findings that exogenous DSBs introduced in mid-pachytene do not affect the number of crossovers in wildtype animals (*Yokoo et al., 2012*).

Next, we examined crossover formation 24 hours post-irradiation (*Figure 2E and F*). For this time-point, we also detected a significant loss of crossover assurance in *pch-2* mutants compared to control worms. However, in contrast to the 8 hour timepoint, we also saw a significant increase in nuclei containing less than six COSA-1 foci in both control (13%) and *pch-2* mutants (26%) compared to unirradiated worms (p-values = 0.0008 and <0.0001, respectively, Fischer's exact test), indicating that even in control animals, extra DSBs in early meiotic prophase can disrupt crossover assurance. In addition, we also observed a significant increase in nuclei with more than six COSA-1 foci in *pch-2* mutants (wildtype: 2%, *pch-2*: 13%), demonstrating that loss of PCH-2 leads to increased crossover formation when nuclei in the transition zone get more DSBs. In other words, PCH-2 inhibits exogenous DSBs introduced in early meiotic prophase from becoming crossovers, supporting our recombination analysis (*Figure 1*).

## PCH-2 prevents SPO-11-induced DSBs from becoming crossovers in early meiotic prophase

Our previous experiment indicated that PCH-2 is preventing exogenous DSBs in early meiotic prophase from becoming crossovers (*Figure 2*). To test whether this was also the case for programmed meiotic DSB formation, we used the auxin-inducible degradation (AID) system to remove the enzyme that is responsible for programmed meiotic DSBs, SPO-11 (*spo-11::AID::3XFLAG*) (*Dernburg et al., 1998*; *Keeney et al., 1997*). When we treated both control and *pch-2* mutant worms for 36 hours with auxin (*Figure 3A*), we observed a complete loss of GFP::COSA-1 foci (*Figure 3C*). When these animals were treated with auxin for 48 hours, to allow nuclei without SPO-11 an additional 12 hours to reach diakinesis (*Figure 3D*), we observed 12 DAPI-stained bodies, or univalents, which are the 6 homolog pairs that have not formed chiasmata (*Figure 3F*). Thus, we can reliably remove SPO-11 from meiotic nuclei throughout the germline and prevent crossover formation.

Since meiotic nuclei travel through the germline at about one cell per row per hour (*Jaramillo-Lambert et al., 2007*), we took advantage of this precise spatiotemporal resolution to test four different timepoints in early meiotic prophase to determine if acute depletion of SPO-11 leads to any changes in crossover formation between control animals and *pch-2* mutants (*Figure 3A and D*). In this experiment, we assayed crossover formation with two experimental approaches. Similar to our irradiation experiment (*Figure 2E and F*), we assayed crossover formation by analyzing the number of GFP::COSA-1 foci in late pachytene 24, 22, 20, and 18 hours after acutely depleting SPO-11 (*Figure 3A*). In addition to this approach, we analyzed the presence of bivalents in diakinesis (*Figure 3D*), which indicates the successful formation of chiasmata between homologous chromosomes. Since meiotic nuclei travel from late pachytene to diakinesis in approximately 12 hours (*Deshong et al., 2014*), the 36, 34, 32, and 30 hour timepoints to assay bivalent formation correspond to the 24, 22, 20, and 18 hour timepoints, respectively, in which GFP::COSA-1 were analyzed. We also performed a control

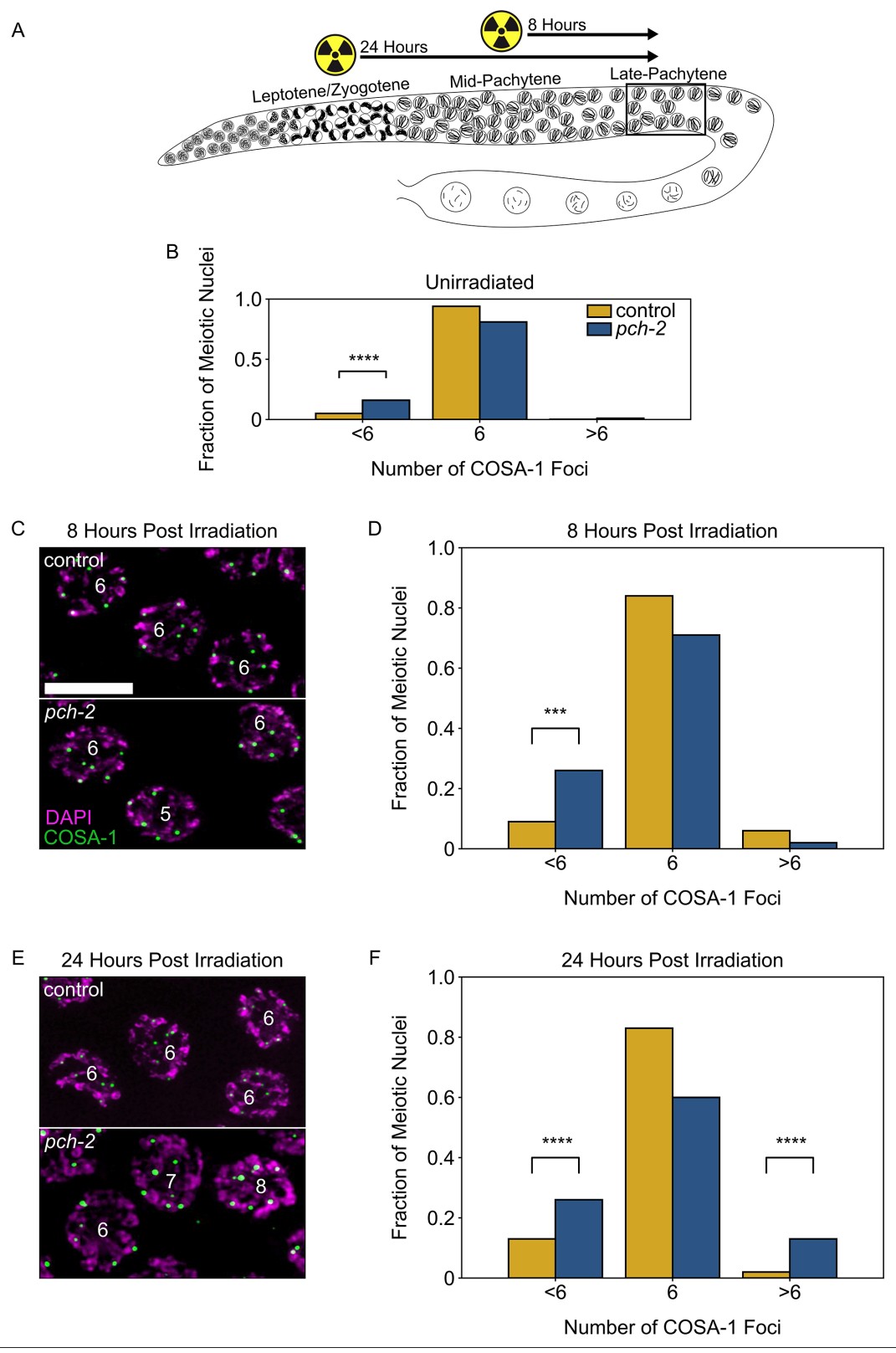

**Figure 2.** PCH-2 prevents exogenous double strand breaks from becoming crossovers in early meiotic prophase. (**A**) Illustration of the irradiation experiments in control and *pch-2* mutants. Box indicates late pachytene, the area where GFP::COSA-1 foci are analyzed. (**B**) Fraction of meiotic nuclei with less than six, six, or greater than six GFP::COSA-1 foci in control animals (yellow, n = 446) and *pch-2* mutants (blue, n = 552). (**C**) Meiotic nuclei in

*Figure 2 continued on next page*

*Figure 2 continued*

control animals and *pch-2* mutants 8 hours post irradiation stained for DAPI (magenta) and GFP::COSA-1 (green). Scale bar is 4 um. (**D**) Fraction of meiotic nuclei with less than six, six, or greater than six GFP::COSA-1 foci in control animals (yellow, n = 143) and *pch-2* mutants (blue, n = 125) 8 hours post irradiation. (**E**) Meiotic nuclei in control animals and *pch-2* mutants 24 hours post irradiation with DAPI (magenta) and GFP::COSA-1 (green). (**F**) Fraction of meiotic nuclei with less than six, six, or greater than six GFP::COSA-1 foci in control animals (n = 179) and *pch-2* mutants (n = 378) 24 hours post irradiation. ***p<0.001, ****p<0.0001.

The online version of this article includes the following source data for figure 2:

**Source data 1.** Numerical data depicted in *Figure 2*.

---

experiment, treating worms with ethanol for 24 and 36 hour timepoints to ensure that this tagged version of SPO-11 was fully functional for crossover formation (*Figure 3C and F*).

At the 24 hour timepoint (*Figure 3A and C*), when meiotic nuclei in the transition zone do not receive SPO-11-induced DSBs, we observe a range of GFP::COSA-1 foci in late pachytene in both control animals and *pch-2* mutants with a majority of nuclei having 0 or 1 focus (control and *pch-2* averages: 1.8 GFP::COSA-1 foci), indicating a loss of crossover formation, albeit with some heterogeneity. This heterogeneity likely reflects that it takes 1–4 hours to significantly affect DSB formation through acute degradation of SPO-11 at this stage of meiosis in *C. elegans* (*Hicks et al., 2022*). In addition, there was no statistically significant difference in the number of GFP::COSA-1 foci between control animals and *pch-2* mutants. However, when we assayed bivalent formation 12 hours later (see 36 hour auxin treatment in *Figure 3D and F*), we observed the same heterogeneity and a slight, statistically significant difference in the number of DAPI-stained bodies between control animals (average number of DAPI-stained bodies: 10.6) and *pch-2* mutants (average number of DAPI-stained bodies: 9.6), suggesting that even at this very early timepoint, we can detect a role for PCH-2 in preventing early SPO-11-induced DSBs from becoming crossovers.

We continued to monitor GFP::COSA-1 and bivalent formation at successive timepoints. At the 22 hour and 34 hour timepoints, where SPO-11 depletion begins later in the transition zone, we saw an increase in the number of GFP::COSA-1 foci (*Figure 3C*) and a reduction in the number of DAPI-stained bodies (*Figure 3F*), suggesting that more DSBs are becoming crossovers. In our analysis of GFP::COSA-1 foci at this timepoint, there appeared to a bimodal distribution: the majority of nuclei in both genotypes fell into two clear classes, those with no COSA-1 foci (24% in control animals, 26% in *pch-2* mutants) and those with six COSA-1 foci (42% in both control animals and *pch-2* mutants) (*Figure 3C*). The number of crossovers continued to increase at the 20 hour and 32 hour timepoints, eventually reaching wildtype numbers for GFP::COSA-1 and DAPI-stained bodies by the 18 hour and 30 hour timepoints, respectively (*Figure 3C and F*).

At the 22 hour and 34 hour timepoints, we did not detect any statistically significant difference in the number of GFP::COSA-1 foci or DAPI-stained bodies between control animals and *pch-2* mutants (*Figure 3C and F*), even when we analyzed the number of nuclei that had greater than six GFP::COSA-1 foci. In contrast, while the number of GFP::COSA-1 foci was not statistically significantly different between control animals and *pch-2* mutants at the 20 hour timepoint (*Figure 3C*), the number of nuclei with greater than six GFP::COSA-1 foci was significantly higher in *pch-2* mutants (control: 3%, *pch-2*: 10%, p=0.009, Fischer's exact test), similar to our analysis of GFP::COSA-1 24 hours after irradiation. Even more strikingly, the number of DAPI-stained bodies was significantly lower in *pch-2* animals than control animals at the corresponding 32 hour timepoint (*Figure 3F*), indicating that more homolog pairs had successfully formed chiasmata in *pch-2* mutants. No difference in GFP::COSA-1 foci or bivalent formation was observed at the 18 hour and 30 hour timepoints, respectively (*Figure 3C and F*). These data indicate that during a narrow time frame of early meiotic prophase, likely during leptotene/zygotene given the delay in limiting DSB formation with auxin-induced degradation of SPO-11, PCH-2 prevents SPO-11-induced DSBs from becoming crossovers and chiasmata.

## PCH-2 is required for timely loading and removal of MSH-5 on meiotic chromosomes through its regulation of HIM-3

After DSB formation, a subset of DSBs are licensed to be repaired through a pro-crossover pathway and eventually winnowed to a stereotypical number of crossovers (*Yokoo et al., 2012*). In *C. elegans*, these crossover-eligible intermediates can be visualized by the loading of the pro-crossover factor

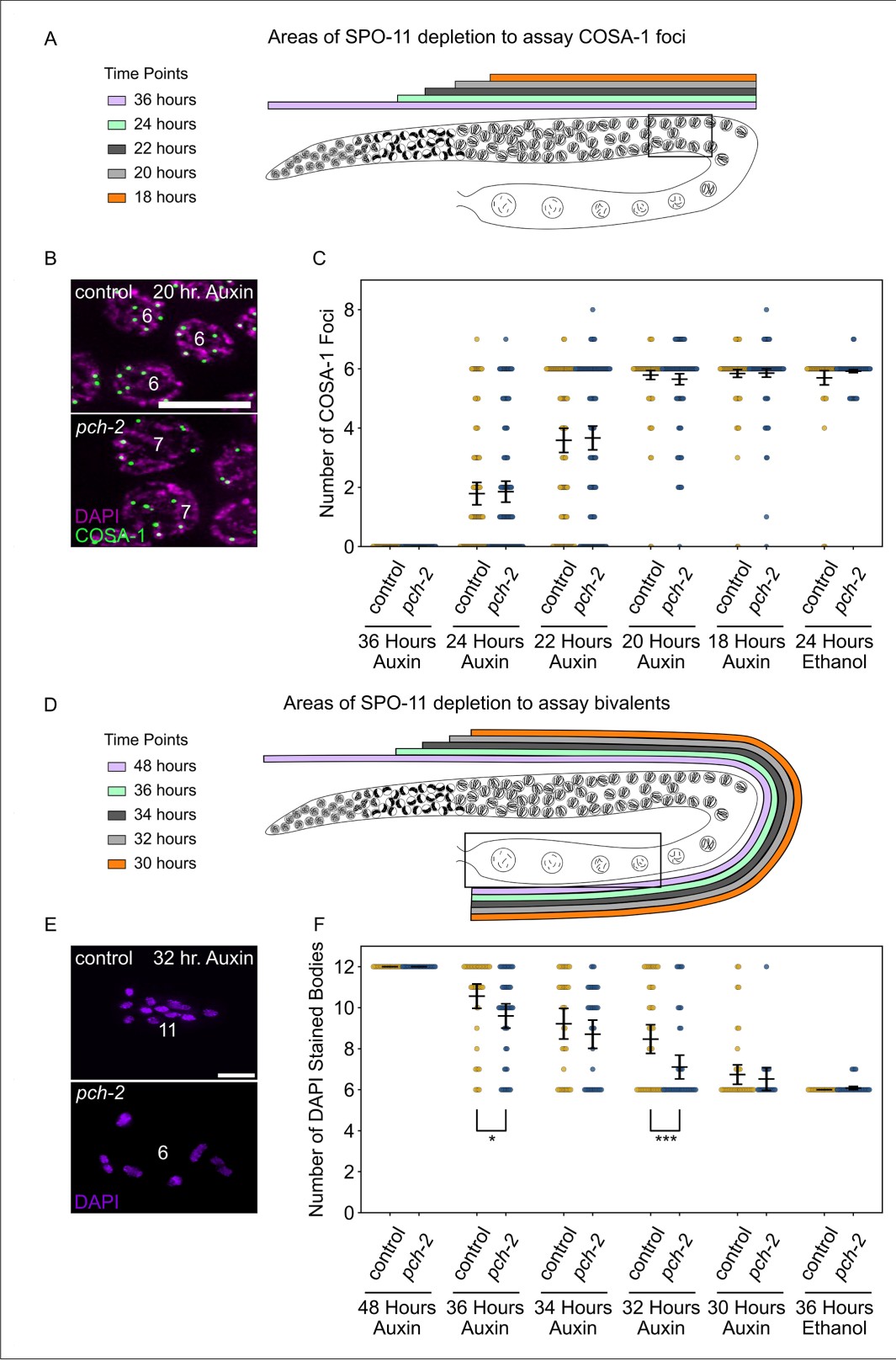

**Figure 3.** PCH-2 prevents SPO-11-induced double-strand breaks from becoming crossovers in early meiotic prophase. (**A**) Illustration of the SPO-11 depletion experiment to assay GFP::COSA-1 in control animals and *pch-2* mutants at different timepoints of auxin treatment. Each timepoint indicates when SPO-11 is depleted in the germline with auxin-induced degradation. Box indicates late pachytene, the area where GFP::COSA-1 foci are

*Figure 3 continued on next page*

*Figure 3 continued*

analyzed. (**B**) Representative images of meiotic nuclei in control animals and *pch-2* mutants treated with auxin for 20 hours, stained for DAPI (magenta) and GFP::COSA-1 (green). Scale bar is 5 um. (**C**) Number of GFP::COSA-1 foci in meiotic nuclei at different timepoints of auxin treatment in control (blue) and *pch-2* mutants (yellow). Error bars represent SEM. N values are as follows: 36 hours on auxin, control (78 nuclei), and *pch-2* (83 nuclei); 24 hours on auxin, control (132 nuclei), and *pch-2* (155 nuclei); 22 hours on auxin, control (152 nuclei), and *pch-2* (168 nuclei); 20 hours on auxin, control (139 nuclei), and *pch-2* (157 nuclei); 18 hours on auxin, control (154 nuclei), and *pch-2* (154 nuclei); and 24 hours on ethanol, control (86 nuclei), and *pch-2* (143 nuclei). (**D**) Illustration of the SPO-11 depletion experiment to assay bivalents in control animals and *pch-2* mutants at different timepoints of auxin treatment. Each timepoint indicates when SPO-11 is depleted in the germline with auxin-induced degradation. Box indicates diakinesis, where DAPI-stained bodies are analyzed. (**E**) Oocytes from control animals and *pch-2* mutants stained for DAPI (magenta). Scale bar is 4 um. (**F**) Number of DAPI-stained bodies in meiotic nuclei at different timepoints of auxin treatment in control animals and *pch-2* mutants. N values are as follows: 48 hours on auxin, control (47 nuclei), and *pch-2* (43 nuclei); 36 hours on auxin, control (46 nuclei), and *pch-2* (50 nuclei); 34 hours on auxin, control (41 nuclei), and *pch-2* (41 nuclei); 32 hours on auxin, control (51 nuclei), and *pch-2* (47 nuclei); 30 hours on auxin, control (46 nuclei), and *pch-2* (21 nuclei); and 36 hours on ethanol, control (52 nuclei), and *pch-2* (48 nuclei). Error bars represent SEM and *<0.05, ***p<0.001.

The online version of this article includes the following source data for figure 3:

**Source data 1.** Numerical data depicted in *Figure 3*.

---

MSH-5, a component of the meiosis-specific MutSy complex that stabilizes crossover-specific DNA repair intermediates called joint molecules (*Janisiw et al., 2018*; *Kelly et al., 2000*; *Snowden et al., 2004*). In the spatiotemporally organized meiotic nuclei of the germline, a functional GFP-tagged version of MSH-5, GFP::MSH-5, begins to form a few foci in leptotene/zygotene (the transition zone), becoming more numerous in early pachytene before decreasing in number in mid pachytene to ultimately colocalize with COSA-1 marked sites in late pachytene in a process called designation (*Janisiw et al., 2018*; *Woglar and Villeneuve, 2018*; *Yokoo et al., 2012*; see control in *Figure 4A and B*). The mechanism through which MSH-5 functions with the SC and other meiosis factors to ultimately form crossovers has not been established but may involve phosphorylation by cyclin-dependent kinases (*Haversat et al., 2022*; *Zhang et al., 2021*).

Given that we have shown that PCH-2 prevents early DSBs from becoming crossovers (*Figures 2 and 3*), we tested whether we could cytologically detect this inhibition at the level of MSH-5 behavior in control animals and *pch-2* mutants. We generated wildtype and *pch-2* mutants with GFP::MSH-5 and quantified the average number of GFP::MSH-5 foci per row of nuclei from the transition zone to late pachytene (*Figure 4A and B*). We observed a statistically significant increase in the average number of GFP::MSH-5 foci per row in the transition zone in *pch-2* mutants (*Figure 4A and B*, p<0.0001, Student's *t*-test), indicating that PCH-2 typically limits MSH-5 loading at this early stage of meiotic prophase. The average number of GFP::MSH-5 foci per row was similar in number in early pachytene in both backgrounds. However, we also observed a substantial increase in the average number of GFP::MSH-5 foci per row in *pch-2* mutants in the mid and late pachytene regions (*Figure 4A and B*), indicating that PCH-2 also promotes the turnover or maturation of GFP::MSH-5 foci at these later stages of meiotic prophase. Thus, PCH-2 prevents the loading of the crossover-promoting factor, MSH-5, limiting the formation of crossover-eligible intermediates during early meiotic prophase, consistent with both our genetic analysis of recombination (*Figure 1*) and PCH-2's role in preventing early DSBs from becoming crossovers (*Figures 2 and 3*). However, PCH-2 also appears to limit the number of crossover-eligible intermediates in mid and late pachytene, an unexpected observation given that the most severe recombination defect in *pch-2* mutants is the loss of crossover assurance (*Deshong et al., 2014*).

We previously showed that PCH-2 genetically interacts with meiotic HORMADs to control different aspects of meiosis. A mutant version of the essential meiotic HORMAD, HIM-3[R93Y], binds its closure motif with reduced affinity in vitro, and we have proposed that an analogous mutation in another essential meiotic HORMAD, HTP-3[H96Y], behaves similarly (*Russo et al., 2023*). These mutations suppress different meiotic defects in *pch-2* mutants in vivo, indicating that PCH-2 regulates pairing and synapsis through its regulation of HTP-3 and crossover recombination through its regulation of HIM-3 (*Russo et al., 2023*). To determine which meiotic HORMAD PCH-2 might be regulating to affect GFP::MSH-5's loading and removal on meiotic chromosomes, and the potential role for these early crossover-eligible

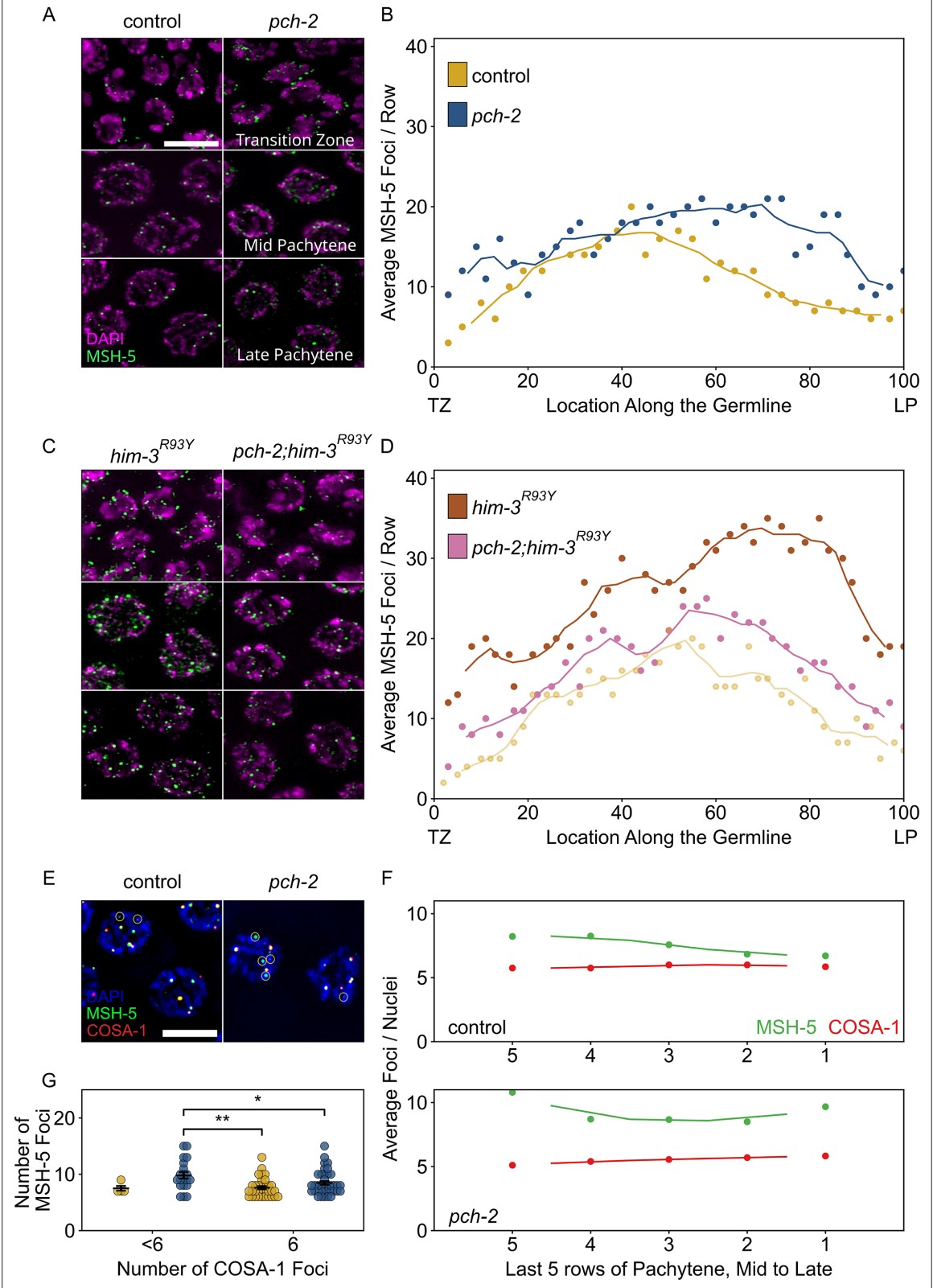

**Figure 4.** PCH-2 is required for timely loading and removal of MSH-5 on meiotic chromosomes through its regulation of HIM-3. (**A**) Representative images of nuclei in different stages of meiotic prophase in control animals and *pch-2* mutants stained for DAPI (magenta) and GFP::MSH-5 (green) Scale bar is 5 um. (**B**) Scatter plot showing average GFP::MSH-5 foci per row of germline nuclei in control animals (yellow, 163 nuclei) and *pch-2* mutants (blue, 195 nuclei) from the transition zone (TZ) to late pachytene (LP), normalized to 100. The line represents a rolling average of four rows. (**C**) Representative

*Figure 4 continued on next page*

Figure 4 continued

images of nuclei in different stages of meiotic prophase in $him-3^{R93Y}$ mutants (left) and $pch-2;him-3^{R93Y}$ double mutants (right), stained for DAPI (magenta) and GFP::MSH-5 (green). (**D**) Scatter plot showing average GFP::MSH-5 foci per row in $him-3^{R93Y}$ (brown, 183 nuclei) and $pch-2;him-3^{R93Y}$ mutants (pink, 163 nuclei) from the TZ to LP, normalized to 100. The line represents a rolling average of four rows. Similar data is provided for a control germline (opaque yellow, 236 nuclei) for comparison. (**E**) Representative images of meiotic nuclei in control animals and $pch-2$ mutants stained for DAPI (blue), GFP::MSH-5 (green), and OLLAS::COSA-1 (red). Yellow circles indicate GFP::MSH-5 without OLLAS::COSA-1. Scale bar is 4 um. (**F**) Scatter plot showing average GFP::MSH-5 (green) and OLLAS::COSA-1 (red) foci per row in the last five rows of the germline in control animals (36 nuclei) and $pch-2$ mutants (45 nuclei). The line represents a rolling average of two rows. (**G**) Swarm plot showing number of GFP::MSH-5 foci in control (yellow, 14 nuclei) and $pch-2$ mutant (blue, 29 nuclei) nuclei with less than six OLLAS::COSA-1 foci (left) and six OLLAS::COSA-1 foci (right). Error bars represent SEM. *$p<0.05$, **$p<0.01$.

The online version of this article includes the following source data and figure supplement(s) for figure 4:

**Source data 1.** Numerical data depicted in *Figure 4* and *Figure 4—figure supplement 1*.

**Figure supplement 1.** PCH-2 does not regulate GFP::MSH-5 loading and removal through HTP-3.

**Figure supplement 2.** *pch-2* meiotic nuclei with elevated numbers of GFP::MSH-5 foci show defects in crossover assurance.

intermediates, we constructed $gfp::msh-5;htp-3^{H96Y}$, $gfp::msh-5;pch-2;htp-3^{H96Y}$, $gfp::msh-5;him-3^{R93Y}$ and $gfp::msh-5;pch-2;him-3^{R93Y}$ mutants and quantified GFP::MSH-5 foci throughout the germline. Both $htp-3^{H96Y}$ and $him-3^{R93Y}$ mutants showed a drastic increase in the average number of GFP::MSH-5 per row of nuclei throughout the germline (*Figure 4C and D*, *Figure 4—figure supplement 1*). For example, compared to control germlines that peaked at approximately an average of 20 MSH-5 foci per row in early pachytene and then decreased in mid-pachytene, $him-3^{R93Y}$ germlines had closer to an average of 30 GFP::MSH-5 foci per row in early pachytene. Moreover, the average number of GFP::MSH-5 foci per row increased further in mid and late pachytene, achieving peaks closer to an average of 35 foci (*Figure 4D*). These very high numbers of GFP::MSH-5 were striking but do not produce an increased number of crossovers in $htp-3^{H96Y}$ or $him-3^{R93Y}$ single mutants (*Russo et al., 2023*), raising the possibility that not all of them reflect functional, crossover-specific intermediates, similar to what has been observed in the absence of synapsis (*Woglar and Villeneuve, 2018*). These data also suggest that these mutations affect the behavior of GFP::MSH-5 on meiotic chromosomes.

Quantification of GFP::MSH-5 foci in $pch-2;htp-3^{H93Y}$ double mutants was similar to $htp-3^{H93Y}$ single mutants, suggesting that PCH-2's effect on the behavior of GFP::MSH-5 foci was not through its regulation of HTP-3 (*Figure 4—figure supplement 1*). In stark contrast to $pch-2;htp-3^{H93Y}$ double mutants, the $pch-2;him-3^{R93Y}$ double mutant showed a dramatic decrease in the overall average of MSH-5 foci per row throughout the germline (*Figure 4C and D*). These averages fell between the averages in control and $pch-2$ mutant animals, particularly in the transition zone and late pachytene (*Figure 4D*), consistent with our previous report that this double mutant has fewer defects in crossover formation than either single mutant (*Russo et al., 2023*). These data indicate that PCH-2's effect on promoting the removal of GFP::MSH-5 in both the transition zone and mid to late pachytene is through its regulation of HIM-3.

$pch-2$ single mutants exhibit a loss of crossover assurance, as visualized by GFP::COSA-1 foci (*Figure 2B* and *Deshong et al., 2014*). However, our quantification of GFP::MSH-5 in $pch-2$ single mutants in mid to late pachytene showed an increased number of crossover-eligible intermediates (*Figure 4A and B*), suggesting a complex relationship between having too many crossover-eligible intermediates and crossover assurance in *C. elegans*. To address this inconsistency, we generated wildtype animals and $pch-2$ mutants with both GFP::MSH-5 and a version of COSA-1 that has been endogenously tagged at the N-terminus with the epitope tag, OLLAS (*Janisiw et al., 2018*), a fusion of the *Escherichia coli* OmpF protein and the mouse Langerin extracellular domain. Consistent with our and others' previous findings, we observe a gradual reduction in the average number of GFP::MSH-5 foci per row of nuclei in control animals as these nuclei approach the end of pachytene, ultimately converging with the average number of OLLAS::COSA-1 foci per row (*Figure 4E and F*; *Janisiw et al., 2018*; *Woglar and Villeneuve, 2018*). In addition, we consistently observed co-localization of GFP::MSH-5 and OLLAS::COSA-1 foci in control animals, with a few MSH-5 foci persisting without COSA-1, consistent with crossover designation (*Figure 4E and F*).

By contrast, we detected higher average numbers of GFP::MSH-5 foci per row across all late pachytene nuclei in $pch-2$ mutants, and this average never converged upon the average number of OLLAS::COSA-1 foci per row (*Figure 4E and F*). To test whether there was a correlation between the

number of GFP::MSH-5 foci and the loss of crossover assurance in *pch-2* mutants, we determined the average number of GFP::MSH-5 foci in meiotic nuclei with six OLLAS::COSA-1 foci and those with less than six OLLAS::COSA-1 foci in both control and *pch-2* mutant animals (*Figure 4E and G*, *Figure 4—figure supplement 2*). Indeed, we observed that *pch-2* mutant nuclei with less than six OLLAS::COSA-1 foci had significantly higher numbers of GFP::MSH-5 foci compared to both control and *pch-2* mutant nuclei with 6 OLLAS::COSA-1 foci (*Figure 4E and G*), suggesting that the inability to reduce the number of crossover-eligible intermediates in *pch-2* mutants, counterintuitively, prevents designation on some chromosomes and contributes to the observed loss of crossover assurance.

## PCH-2 is removed from the synaptonemal complex when crossovers are designated

We have previously shown that PCH-2 localization to the SC is extended when there are partial defects in synapsis or changes in karyotype, producing an increase in crossovers and a loss of crossover interference (*Deshong et al., 2014*; *Patel et al., 2023*). These data led us to hypothesize that PCH-2's presence on chromosomes promotes crossover formation. However, our quantification of GFP::MSH-5 indicates that PCH-2 is required to limit the number of crossover-eligible intermediates, in direct contrast to our proposed hypothesis (*Figure 4*). Therefore, we decided to revisit what role PCH-2 localization to the SC might play in regulating crossover formation. To this end, we localized PCH-2 in *dsb-2* mutants, in which DSB formation is substantially reduced and fewer crossovers are formed in *C. elegans* (*Rosu et al., 2013*). We performed this experiment because, in budding yeast, similar mutants that reduce DSB formation rely on Pch2 to successfully complete meiosis (*Joshi et al., 2009*; *Zanders and Alani, 2009*).

Because the recombination defect in *dsb-2* mutants worsens with age (*Rosu et al., 2013*), we looked at PCH-2 localization in both young (24 hours post L4 larval stage) and older animals (48 hours post L4). We observed a unique and striking localization pattern in *dsb-2* mutants that we had not observed before. When stained for PCH-2 and GFP::COSA-1, nuclei retain PCH-2 onto chromosomes far into late pachytene, past the normal region when PCH-2 typically is removed from chromosomes (*Figure 5A and B*). However, unlike what we have observed in other mutants that have defects in synapsis or changes in karyotype (*Deshong et al., 2014*; *Patel et al., 2023*), this retention of PCH-2 is not uniform among all late pachytene nuclei in *dsb-2* mutants. Instead, most nuclei lose PCH-2 localization in late pachytene while some retain it. We tested whether there was a relationship between the retention of PCH-2 and the number of GFP::COSA-1 foci and found that 96% of nuclei that lose PCH-2 have at least one GFP::COSA-1 focus (*Figure 5A and B*). In older animals (see 48 hours post-L4), this pattern was even more clear: 77% of nuclei without PCH-2 had one or more GFP::COSA-1 focus and 93% of meiotic nuclei without a GFP::COSA-1 focus retained PCH-2.

These data indicate that PCH-2's removal from the SC is in response to, coincident with or facilitates crossover designation. To distinguish between these possibilities, we analyzed GFP::COSA-1 foci in *dsb-2::AID* (*Zhang et al., 2018*) and *dsb-2::AID;pch-2* worms, reasoning that if PCH-2's removal is in response to or coincident with designation, we should not detect any differences in the number of GFP::COSA-1 foci in *dsb-2::AID* and *dsb-2::AID;pch-2* worms. We performed these experiments with the AID system because of variability in the *dsb-2* mutant background that affected reproducibility of our experiments.

Upon auxin treatment of *dsb-2::AID* worms, we observed a statistically significant (p<0.0001, Mann–Whitney *U* test) decrease in the number of GFP::COSA-1 foci, in comparison with ethanol-treated worms, verifying that we can reliably knock down DSB-2 with the AID system (*Figure 5D*). When we performed the same experiment in *dsb-2::AID;pch-2* worms, the average number of GFP::COSA-1 foci further decreased, indicating that fewer crossovers are designated when DSBs are reduced and PCH-2 is absent (*Figure 5C and D*). These data argue against the possibility that PCH-2's removal from the SC is simply in response to or coincident with crossover designation and instead suggest that PCH-2's removal from the SC somehow facilitates crossover designation and assurance. Given the correlation between elevated GFP::MSH-5 foci and the loss of crossover assurance we observe in *pch-2* mutants, we propose that PCH-2 is retained on meiotic chromosomes to ensure that extra crossover-eligible intermediates are removed and crossover designation is delayed until crossover assurance can be guaranteed in *C. elegans*.

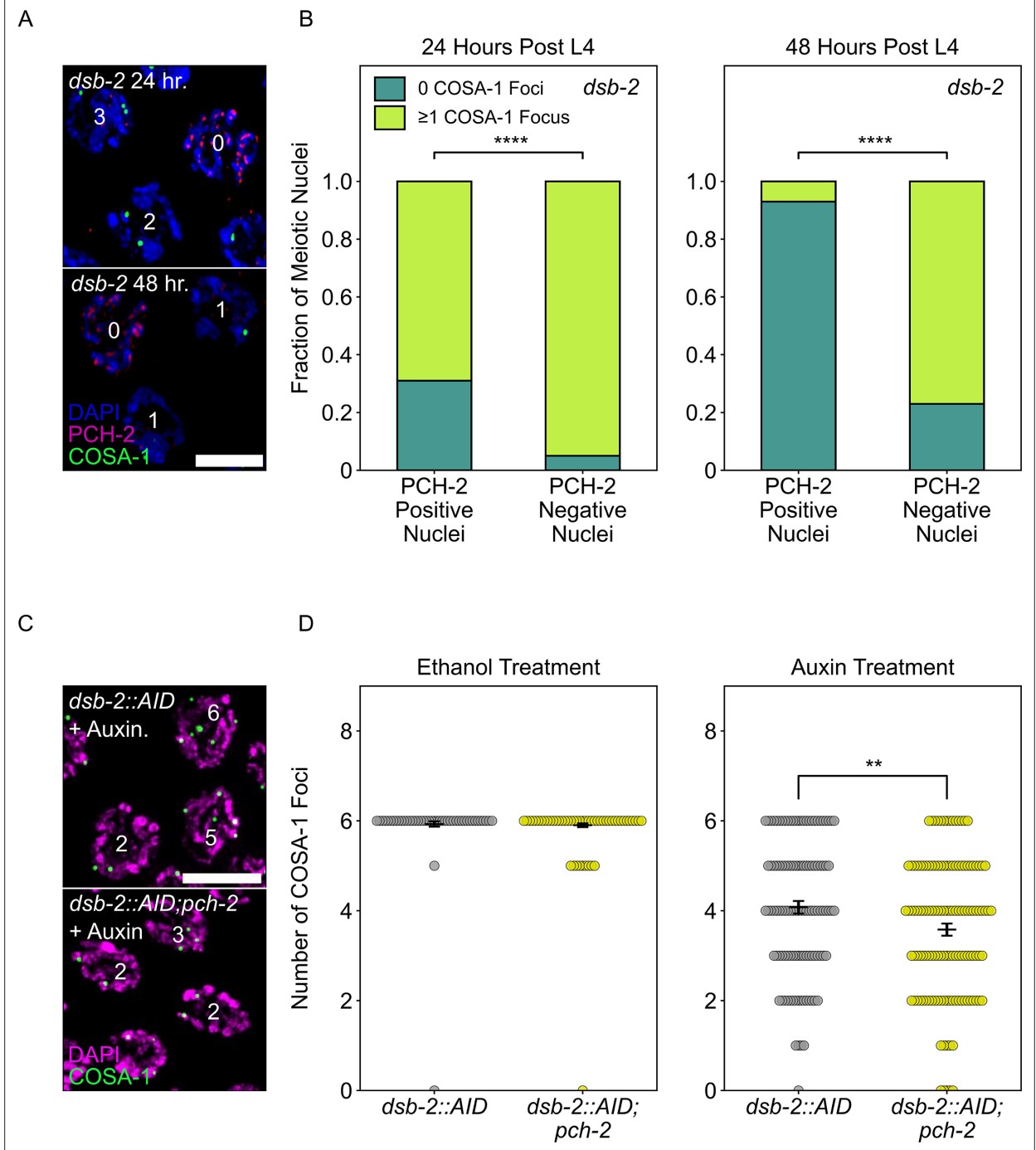

**Figure 5.** PCH-2 is removed when crossovers are designated. (**A**) Representative images of meiotic nuclei in *dsb-2* animals 24 hours post L4 and 48 hours post L4 stained for DAPI (magenta), PCH-2 (red), and GFP::COSA-1 (green). Scale bar is 4 um. (**B**) Stacked histograms showing percentage of PCH-2 positive (n = 43 at 24 hours, n = 42 at 48 hours) and negative (n = 194 at 24 hours, n = 130 at 48 hours) nuclei with (lime) and without (dark green) GFP::COSA-1 foci in *dsb-2* mutants at 24 hours post L4 and 48 hours post L4. (**C**) Representative images of meiotic nuclei in *dsb-2::AID* and *dsb-2::AID;pch-2* mutants treated with auxin and stained for DAPI (magenta) and GFP::COSA-1 (green). Scale bar is 5 um. (**D**) Swarm plot showing the number of GFP::COSA-1 foci in *dsb-2::AID* (gray) and *dsb-2::AID;pch-2* (lemon) mutants when treated with ethanol or auxin. N values are as follows: *dsb-2::AID* (101 nuclei) and *dsb-2::AID;pch-2* (154 nuclei) on ethanol, *dsb-2::AID* (136 nuclei) and *dsb-2::AID;pch-2* (131 nuclei) on auxin. Error bars represent the SEM. \*\*p<0.01, \*\*\*\*p<0.0001.

The online version of this article includes the following source data for figure 5:

**Source data 1.** Numerical data depicted in *Figure 5*.

# PCH-2 and high CHK-2 activity control the fate of early double-strand breaks

In *C. elegans*, meiotic cell cycle entry and progression depends on the activity of CHK-2, the meiosis-specific ortholog of the DNA-damage kinase Chk2/CHEK2 (*Baudrimont et al., 2022*; *Castellano-Pozo et al., 2020*; *Kim et al., 2015*; *MacQueen and Villeneuve, 2001*; *Zhang et al., 2023*). Meiotic nuclei in leptotene/zygotene are characterized by high CHK-2 activity, which drops to intermediate activity in mid-pachytene (*Kim et al., 2015*; *Zhang et al., 2023*). In late pachytene, CHK-2 activity is inactivated by the recruitment of polo-like kinase, PLK-2, to the SC, which enables crossover designation (*Zhang et al., 2023*). Thus, CHK-2 activity also has consequences for the progression of meiotic recombination and DNA repair.

Given that early DSBs are prevented from becoming crossovers by PCH-2 in early meiotic prophase, we wanted to test if high CHK-2 activity in leptotene/zygotene also contributed to the fate of these early DSBs. To evaluate this possibility, we used *syp-1^T452A* mutants (*Sato-Carlton et al., 2018*). The SC component, SYP-1, is phosphorylated by cell cycle kinase, CDK-1, on T452, producing a Polo box binding motif that recruits the polo-like kinase, PLK-2, to the SC, contributing to the inactivation of CHK-2 when chromosomes are synapsed (*Brandt et al., 2020*; *Zhang et al., 2023*). Therefore, in *syp-1^T452A* mutants, CHK-2 activity remains high throughout most of meiotic prophase in the *C. elegans* germline, delaying meiotic progression as visualized by the extension of the transition zone (*Figure 6A*, *Figure 6—figure supplement 1*).

We quantified the total number of GFP::COSA-1 foci in late pachytene nuclei in *syp-1^T452A* mutants and observed a significant decrease in the average number of GFP::COSA-1 foci compared to control animals (*Figure 6B and C*), consistent with previous findings that *syp-1^T452A* mutants delay designation and have fewer crossovers (*Zhang et al., 2023*). These data are also consistent with the possibility that some DSBs fail to become crossovers when CHK-2 activity remains high. If this hypothesis is correct, we predict that the combination of high CHK-2 activity and loss of *pch-2* should produce more crossovers. We generated *pch-2;syp-1^T452A* double mutants to test this hypothesis. These double mutants exhibited the same delay in meiotic progression as *syp-1^T452A* single mutants (*Figure 6—figure supplement 1*). When we quantified GFP::COSA-1 in *pch-2;syp-1^T452A* double mutants, we detected a significant increase in the average number of GFP::COSA-1 foci compared to *syp-1^T452A* single mutants (*Figure 6C*), in strong support of our hypothesis.

To further verify that the *pch-2* mutation suppresses the crossover defect in *syp-1^T452A* mutants, we also monitored bivalent formation in diakinesis nuclei. *syp-1^T452A* single mutants exhibit an average of 7.12 DAPI staining bodies in diakinesis, higher than both control animals and *pch-2* single mutants (*Figure 6D*). By contrast, *pch-2;syp-1^T452A* mutants had an average of 6.05 DAPI staining bodies (*Figure 6D and E*), directly supporting our COSA-1 analysis and indicating that both PCH-2 function and high CHK-2 activity collaborate to control the fate of DSBs and prevent some of them from becoming crossovers in early meiotic prophase.

## Discussion

We have shown that PCH-2 antagonizes crossover formation throughout meiotic prophase (*Figures 2–4*) and that this regulation occurs through one of the three essential meiotic HORMADs in *C. elegans*, HIM-3 (*Figure 4*). We propose that PCH-2 remodels HIM-3 on meiotic chromosomes to destabilize crossover-eligible intermediates, visualized in our experiments as GFP-MSH-5 foci, thus limiting which DSBs will become crossovers (*Figure 7*). However, this antagonism has different consequences depending on when during meiotic prophase it occurs, underscoring the importance of temporal regulation of these events. During leptotene/zygotene, when CHK-2 activity is high (*Kim et al., 2015*; *Zhang et al., 2023*) and PCH-2 is present as foci on chromosomes (*Deshong et al., 2014*), PCH-2 prevents crossover formation at some sites of initial DSB formation and early homolog interactions (*Figure 7B*). In this way, PCH-2 promotes a wider distribution of crossovers across the genome (*Figure 1*). In pachytene, when CHK-2 activity has decreased (*Kim et al., 2015*; *Zhang et al., 2023*) and PCH-2 is localized to the SC (*Deshong et al., 2014*), PCH-2 winnows crossover-eligible intermediates marked by GFP-MSH-5, ensuring their designation, colocalization with COSA-1 and crossover assurance (*Figure 7B*). When there are defects in recombination, such as partial synapsis (*Deshong et al., 2014*), changes in karyotype (*Patel et al., 2023*), or too few DSBs (*Figure 5*), PCH-2 persists

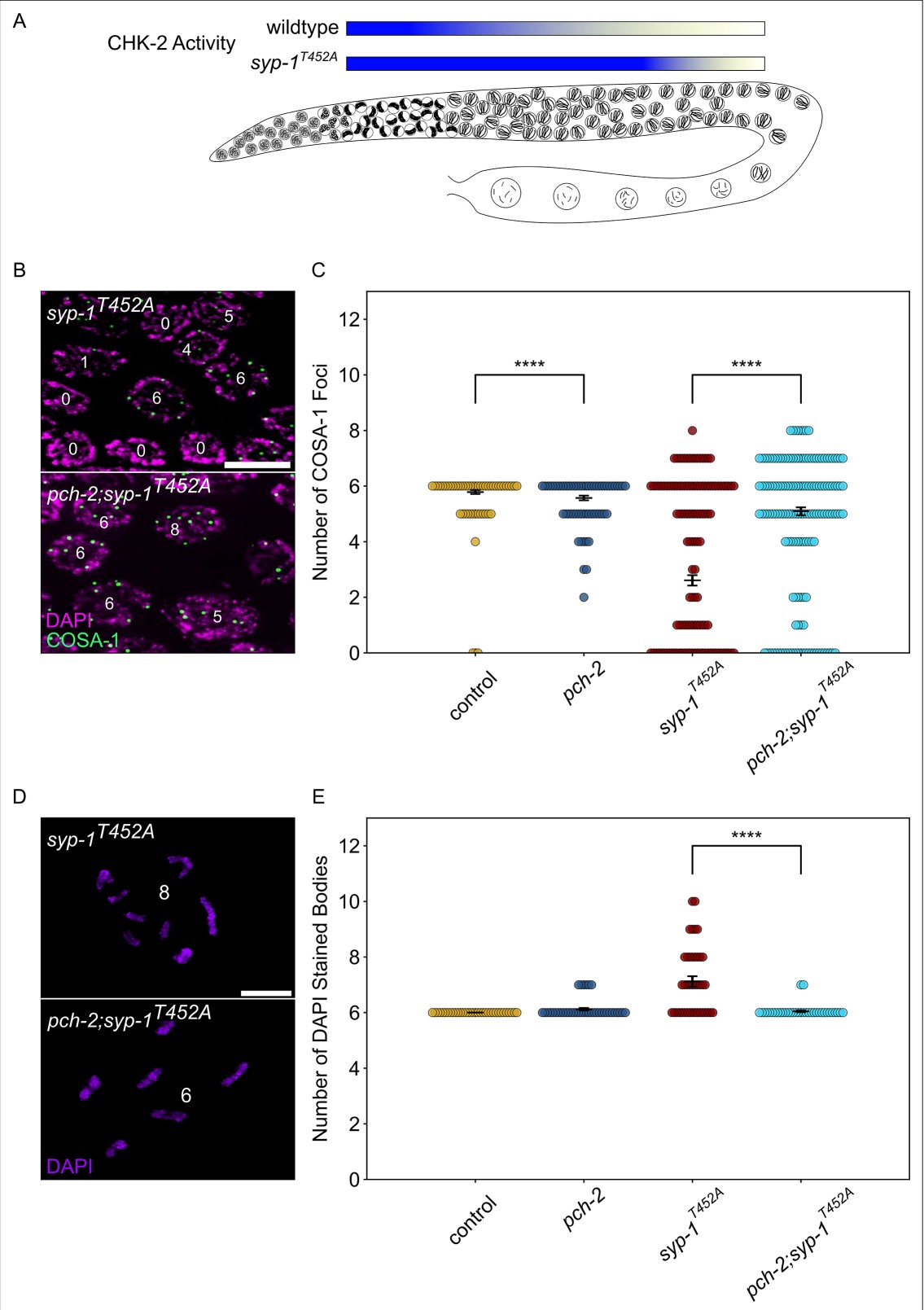

**Figure 6.** PCH-2 and high CHK-2 activity control the fate of early double-strand breaks. (**A**) Illustration of CHK-2 activity in wildtype and *syp-1^{T452A}* germlines. (**B**) Representative images of meiotic nuclei late pachytene in *syp-1^{T452A}* and *pch-2;syp-1^{T452A}* mutants stained for DAPI (magenta) and GFP::COSA-1 (green). Scale bar is 5 um. (**C**) Swarm plot showing number of GFP::COSA-1 foci in control animals (blue), *pch-2* (yellow), *syp-1^{T452A}* (maroon), and *pch-2;syp-1^{T452A}* (light blue) mutants. Error bars represent SEM. (**D**) Oocytes from *syp-1^{T452A}* and *pch-2;syp-1^{T452A}* mutant worms stained for

*Figure 6 continued on next page*

*Figure 6 continued*

DAPI (magenta). Scale bar is 4 um. (**D**) Swarm plot showing number of DAPI-stained bodies in control animals (blue, n = 154), *pch-2* (yellow, n = 89), *syp-1^T452A* (maroon, n = 247), and *pch-2;syp-1^T452A* (light blue, n = 242) mutants. Error bars represent SEM. \*\*\*\*p<0.0001.

The online version of this article includes the following source data and figure supplement(s) for figure 6:

**Source data 1.** Numerical data depicted in *Figure 6*, *Figure 6—figure supplement 1*.

**Figure supplement 1.** *syp-1^T452A* and *pch-2;syp-1^T452A* mutants display a similar defect in meiotic progression.

on the SC to prevent designation, guarantee crossover assurance and some degree of homeostasis, independent of an additional feedback mechanism that increases DSB formation (*Patel et al., 2023*). That persistence of PCH-2 also disrupts crossover interference in two of these situations (*Deshong et al., 2014*; *Patel et al., 2023*) strongly suggests that crossover interference is mechanistically linked to assurance and homeostasis. For example, PCH-2's persistence on the SC may maintain an extended period of competency for interhomolog repair, as has been observed in mutants defective in crossover recombination (*Rosu et al., 2011*).

*him-3^R93Y*;*pch-2* double mutants have stronger crossover assurance than either single mutant but less than wildtype animals (*Russo et al., 2023*), a phenotype which can now be explained by the behavior of GFP::MSH-5 foci in these double mutants (*Figure 4C and D*). We have shown that HIM-3^R93Y mutant protein can adopt the closed conformation and loads on meiotic chromosomes similar to wildtype HIM-3 (*Russo et al., 2023*). In vitro analysis shows that HIM-3^R93Y binds its closure motif with reduced affinity, likely affecting its ability to adopt the closed conformation in vivo (*Figure 7A*; *Russo et al., 2023*). The proposed role of PCH-2 in meiosis is to remodel meiotic HORMADs from the closed conformation to the extended one (*Figure 7A*), regulating their association with chromosomes (*Bhalla, 2023*). However, since meiotic HORMADs are not visibly depleted from chromosomes during meiotic progression in *C. elegans* (*Couteau et al., 2004*; *Couteau and Zetka, 2005*; *Goodyer et al., 2008*; *Martinez-Perez and Villeneuve, 2005*), this genetic interaction supports a role for PCH-2 in temporarily reducing the occupancy of meiotic HORMADs on meiotic chromosomes, destabilizing interactions with partner proteins that modulate the progression and fidelity of meiotic recombination (*Russo et al., 2023*). Thus, PCH-2's remodeling of HIM-3 would disrupt protein–protein interactions that underlie crossover-eligible intermediates, destabilizing them and reducing their number on chromosomes, contributing to crossover control (*Figure 7B*). Since meiotic HORMADs are essential for meiotic chromosome axis structure and function, our data therefore support a role for the meiotic axis in crossover control, as suggested by previous reports (*Chu et al., 2024*; *Girard et al., 2023*; *Lambing et al., 2020*; *Nabeshima et al., 2004*). However, the disassembly of crossover-eligible intermediates would also release pro-crossover factors present at these sites, such as MSH-5, to concentrate at other, more stable sites on the SC, facilitating designation during synapsis (*Girard et al., 2023*; *Yokoo et al., 2012*). In this way, PCH-2's mechanism of action may reconcile current competing models of crossover control (*Girard et al., 2023*). Moreover, these results raise the possibility that this behavior of PCH-2 and/or meiotic HORMADs might be regulated by post-translational modifications associated with crossover control, such as ubiquitination, SUMOylation, and phosphorylation (*Gray and Cohen, 2016*).

Given that mutation of PCH-2 produces changes in the number and distribution of crossovers across multiple model systems, we argue that controlling crossover distribution and number is the conserved role of PCH-2 in meiosis (*Bhalla, 2023*). We have previously proposed that the conserved role of PCH-2 is to coordinate meiotic recombination with synapsis, explaining its role in the pachytene checkpoint (*Bhalla, 2023*). With these results, we further refine this model. Early in meiotic prophase, PCH-2 remodels meiotic HORMADs to prevent some DSBs from becoming crossover-eligible intermediates, widening the recombination landscape beyond early homolog interactions and/or sites that tend to be more favorable for DSB formation, also known as 'hot spots'. For example, this may explain the localization of TRIP13, the PCH-2 ortholog in mice, to telomeres (*Chotiner et al., 2024*), sites that experience early homolog interactions due to the organization of meiotic chromosomes in the bouquet formation (*Scherthan et al., 1996*). This antagonism may also expand the regions of the genome that initiate synapsis in organisms that use DSBs to accomplish this event. This possibility is supported by the observation that loss of TRIP13 in mammals produce meiotic chromosomes that exhibit partial asynapsis (*Roig et al., 2010*), particularly near regions that may act as barriers to

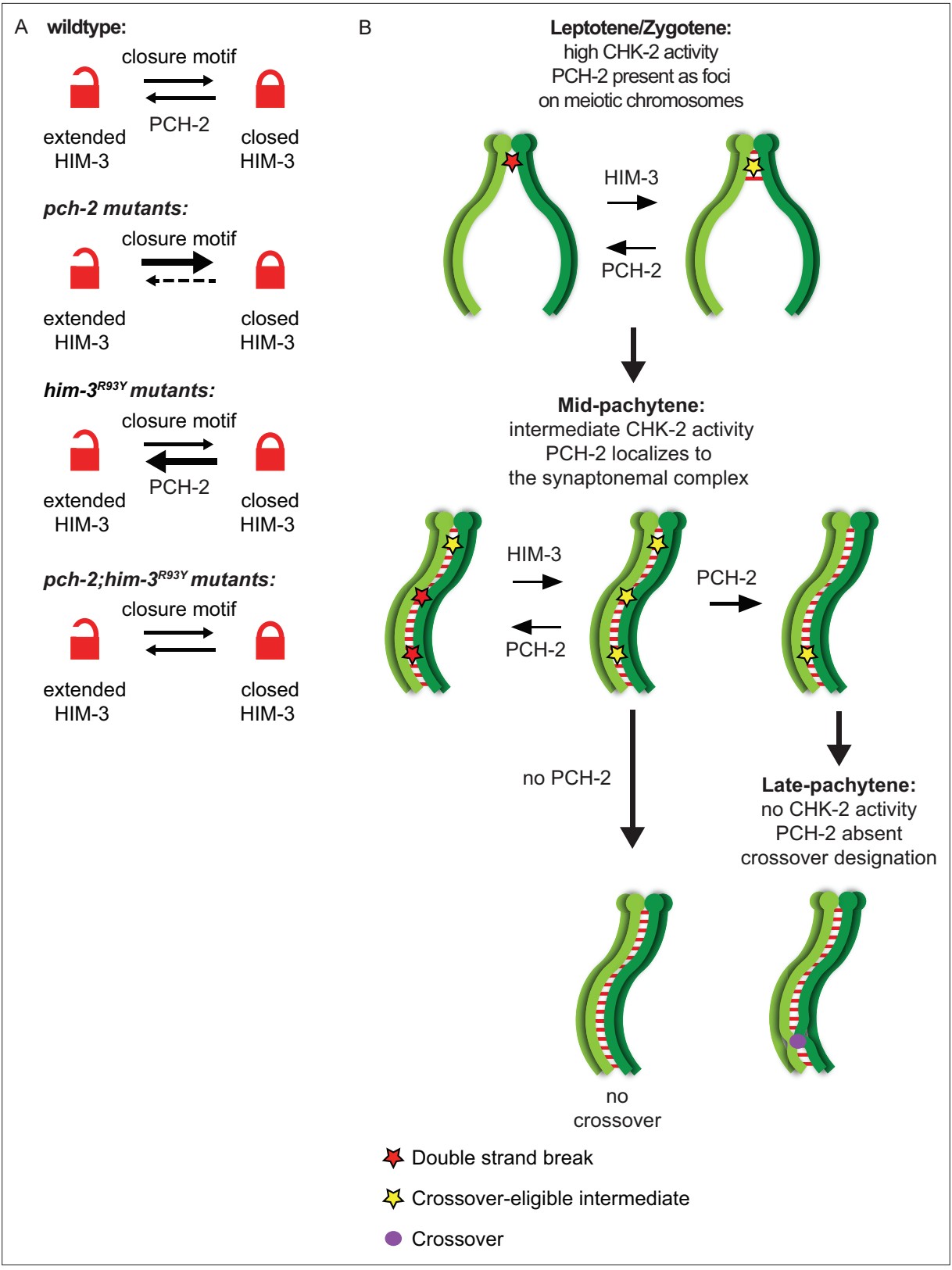

**Figure 7.** PCH-2 remodels HIM-3 to disassemble crossover-eligible intermediates, controlling crossover distribution and number. (**A**) Model for how *pch-2* and *him-3*^R93Y mutations genetically interact to affect the progression of meiotic recombination. HIM-3 adopts the closed conformation upon binding an interacting protein with a closure motif and its conversion to the extended conformation is facilitated by PCH-2's remodeling of its HORMA domain. (**B**) Model for how PCH-2 and HIM-3 progressively implement meiotic recombination during different stages of meiotic prophase.

SC polymerization (*Brown et al., 2005*; *Roig et al., 2010*). Limiting which DSBs become crossover-eligible intermediates in early meiotic prophase also ensures that meiotic recombination overlaps with synapsis, either completely, as in *C. elegans* (*Yokoo et al., 2012*) or partially, as in budding yeast, plants and mice (*Capilla-Pérez et al., 2021*; *Cole et al., 2012*; *Joshi et al., 2015*; *Morgan et al., 2021*). Once synapsis is complete, PCH-2 continues to remodel meiotic HORMADs on chromosomes to control the gradual implementation of crossover number and distribution, reinforcing the important role that synapsis plays in mediating crossover control (*Durand et al., 2022*; *Libuda et al., 2013*).

Unexpectedly, the inability to reduce the number of crossover-eligible intermediates in *pch-2* mutants, as visualized by GFP::MSH-5 foci, does not produce extra crossovers but a loss of crossover assurance in *C. elegans* (*Figure 4*), a somewhat counterintuitive result. One interpretation of these data is that crossover-eligible intermediates may be more numerous but absent from some chromosomes in *pch-2* mutants, explaining the loss of crossover assurance. Since the absence of crossover intermediates in *C. elegans* is accompanied by premature desynapsis of individual chromosomes (*Machovina et al., 2016*; *Pattabiraman et al., 2017*) and chromosomes in *pch-2* mutants delay desynapsis (*Deshong et al., 2014*), we do not favor this interpretation. Instead, we propose that having too many crossover-eligible intermediates can be as deleterious to crossover assurance as having too few (*Figure 7B*). This possibility is further supported by the loss of crossover assurance we detect in irradiated wildtype worms, which is exacerbated in *pch-2* mutants (*Figure 2*).

This phenomenon, where crossover-eligible intermediates need to be winnowed to some threshold number to ensure crossover assurance, may explain the loss of crossover assurance also observed in *Trip13*-deficient mice (*Roig et al., 2010*) and on small chromosomes in budding yeast (*Chakraborty et al., 2017*). Alternatively, the counterintuitive relationship between the number of crossover-eligible precursors and crossover assurance in *pch-2* mutants we observe might reflect an additional layer of regulation during crossover formation specific to *C. elegans*. Since *C. elegans* chromosomes are holocentric, crossovers play an additional role organizing chromosomes for the ordered release of sister chromatid cohesion during meiosis I (*Martinez-Perez et al., 2008*; *Nabeshima et al., 2005*) and extra crossovers can be deleterious to accurate chromosome segregation (*Hollis et al., 2020*). By contrast, in *Arabidopsis*, a system that appears to be able to tolerate an extraordinarily high number of crossovers with little to no effect on chromosome segregation (*Durand et al., 2022*), PCH2's inability to localize to the SC produces an increase in crossover formation, as visualized by both MLH1 foci and the formation of chiasmata (*Yang et al., 2022*). Once again, an overarching theme that becomes apparent in our model is that PCH-2 may play a common role in different systems, with dramatic variations in phenotypic consequences given species-specific requirements and constraints.

We were not surprised to see high numbers of double crossovers on almost every chromosome in our genetic analysis of recombination in wildtype worms, given our previous analysis (*Deshong et al., 2014*). However, we were surprised to see that the majority of them were found near PCs and sites of synapsis initiation, suggesting a relationship between early homolog interactions and the formation of double crossovers. When we revisited our previous data, we observed similar patterns. In addition, we do not detect these double crossovers cytologically in *C. elegans*, even when using the OLLAS::COSA-1 reporter, which has been reported to identify double crossovers in spermatogenesis not visualized by GFP::COSA-1 (*Cahoon et al., 2023*). Crossovers that are cytologically marked by COSA-1 are known as class I crossovers, which depend on pro-crossover factors such as MSH-5 and ZHP-3, rely on synapsis, and exhibit crossover control (*Gray and Cohen, 2016*). These data raise the intriguing possibility that these double crossovers are the product of the alternate, class II, pathway of crossover formation, which relies on a different suite of proteins, does not respond to crossover control, does not depend on synapsis, and contributes to varying degrees in different model systems (*Gray and Cohen, 2016*; *Youds et al., 2010*). Thus, based on the close, functional relationship that exists between class I crossovers and synapsis and the apparent antagonistic relationship that exists between class II crossovers and synapsis, important corollaries of our model may be that PCH-2 specifically coordinates recombination with synapsis to promote class I crossovers, limit class II crossovers and that class II crossovers are more likely to form early in meiosis, prior to synapsis. Therefore, variations in the contribution of the class II crossover pathways to crossover recombination and the degree of cross-talk between class I and II pathways among model systems might reflect the degree to which crossover formation overlaps with synapsis (*Bhalla, 2023*; *Gray and Cohen, 2016*; *Yokoo et al., 2012*). Furthermore, these corollaries are entirely consistent with PCH-2's absence from the

genome of fission yeast and Tetrahymena (*Kops et al., 2020*; *Wu and Burgess, 2006*), both systems in which chromosomes do not undergo meiotic synapsis, crossovers do not exhibit interference and all crossovers are dependent on the class II pathway (*Hollingsworth and Brill, 2004*; *Lukaszewicz et al., 2013*; *Wolfe et al., 1976*). However, this aspect of the model needs to be formally tested.

Our results also have some additional important implications about the regulation of DSB formation across the genome in *C. elegans*. It is formally possible that PCH-2 controls DSB distribution. However, the complexity of recombination defects in *pch-2* mutants argues against this possibility. Instead, we explain the shift in the recombination landscape away from the central regions of chromosomes and toward PC ends in *pch-2* mutants (*Figure 1*) as a result of early DSBs becoming crossovers at the expense of later DSBs. This explanation is better supported by the panel of defects observed in *pch-2* mutants: DSBs introduced early in meiosis become crossover eligible intermediates and crossovers (*Figures 2–4*) and while the number of DSBs is constant, DNA repair is accelerated (*Deshong et al., 2014*). Moreover, this explanation suggests that when DSBs happen in meiotic prophase affects where they happen in the genome. Specifically, we propose that chromosome arms, which are gene poor, receive DSBs early during (or even throughout) DSB formation and the center of chromosomes, which are gene rich, receive DSBs later. The shift in recombination to the center of chromosomes when defects in meiosis prolong DSB formation provides further support to this possibility (*Carlton et al., 2006*; *Deshong et al., 2014*). A similar regulation of the timing of DSB formation has been demonstrated in budding yeast, where the DSB landscape across the whole genome expands when time in prophase is increased (*López Ruiz et al., 2024*) and small, highly recombinogenic, chromosomes, get more DSBs later in meiotic prophase (*Murakami et al., 2020*; *Subramanian et al., 2019*). However, this temporal regulation has not been reported previously in *C. elegans* and suggests that this phenomenon is more widely conserved. This expansion of the DSB landscape in *C. elegans* to include the center regions of chromosomes later in meiosis may be a deliberate attempt for recombination to create new haplotypes for evolution to act on, despite the relative paucity of DSBs and the observation that they can result in chromosome missegregation (*Altendorfer et al., 2020*).

Finally, our work raises some important questions about the functional role(s) of DSBs in meiosis, aside from their contributions to crossover formation. Hicks and colleagues were the first to report that early DSBs do not contribute to crossover formation in *C. elegans* (*Hicks et al., 2022*). Here we show that these early DSBs are prevented from becoming crossovers by both PCH-2 activity and cell cycle stage, specifically in leptotene/zygotene, when homologs are initiating pairing and synapsis. In budding yeast, similar, early-occurring DSBs have been characterized as 'scout DSBs' because of their preference for repair from sister chromatids, versus homologous chromosomes, and their proposed role in contributing to homolog pairing (*Borde and de Massy, 2015*; *Joshi et al., 2015*). The homolog bias that these 'scout DSBs' do display seems dependent on budding yeast *PCH2* (*Joshi et al., 2015*) but interpreting this experiment is complicated by the fact that Pch2 is also required to make the budding yeast meiotic HORMAD, Hop1, available for its loading onto meiotic chromosomes (*Herruzo et al., 2021*).

In contrast to budding yeast, *C. elegans* does not rely on DSBs to promote homolog pairing and initiate synapsis (*Dernburg et al., 1998*); in worms, *cis*-acting sites called PCs are essential for homolog pairing and synapsis (*MacQueen et al., 2005*). There is certainly some support in the literature for DSBs playing a role in supporting pairing and synapsis in *C. elegans* (*Guo et al., 2022*; *Mlynarczyk-Evans et al., 2013*; *Roelens et al., 2015*). However, if these early DSBs were contributing to pairing and synapsis, we would expect to see a genetic interaction between *htp-3*[H96Y] and *pch-2* mutations in the installation of GFP::MSH-5 in the transition zone; we previously reported that *htp-3*[H96Y] suppresses the acceleration of pairing and synapsis of *pch-2* mutants, particularly when PC function is compromised (*Russo et al., 2023*). Instead, we favor the possibility that these early DSBs are generated to amplify the signaling of DNA damage kinases to prime them for their role in recombination, a role that has also been proposed for 'scout' DSBs (*Joshi et al., 2015*). That ATM-1, a conserved DNA damage kinase that is downstream of CHK-2 in *C. elegans*, relies on DSBs for full activity, supports this proposed, conserved role (*Yu et al., 2023*).

Our work provides an important framework to finally understand the role of PCH-2 in controlling the number and distribution of crossovers, a role that we argue is its conserved role. While specific details may vary across systems, we propose that PCH-2 remodels meiotic HORMADs throughout meiotic prophase to destabilize crossover-eligible precursors, coordinating meiotic recombination

with synapsis, contributing to the progressive implementation of meiotic recombination, guaranteeing crossover assurance, interference and homeostasis, and explaining its function as a pachytene checkpoint component.

## Materials and methods

### *C. elegans* genetics and genome engineering

The *C. elegans* Bristol N2 was used as the wildtype strain (*Brenner, 1974*). All strains were maintained at 20°C under standard conditions unless stated. Mutant combinations were generated by crossing. The following mutants and rearrangements were used:

> LGII: *pch-2(tm1458)*, *meIs8* ([*pie-1p::GFP::cosa-1 + unc-119(+)]*), *dsb-2(me96)*, *dsb-2(ie58[dsb-2::AID::3xFLAG])*
> LGIII: *cosa-1(ddr12[OLLAS::cosa-1])*; *htp-3(vc75)*
> LGIV: *him-3(blt9)*, *spo-11(ie59[spo-11::AID::3xFLAG])*, *msh-5[ddr22(GFP::msh-5)]*, *ieSi38 [sun-1p::TIR1::mRuby::sun-1 3'UTR + Cbr-unc-119(+)]*, *nT1[qIs51] (IV;V)*
> LGV: *syp-1(icm85[T452A])*, *bcIs39 (Plin-15::ced-1::GFP)*, *nT1[qIs51] (IV;V)*

### Genetic analysis of recombination

The wildtype Hawaiian CB4856 strain (HI) and the Bristol N2 strain were used to assay recombination between SNPs on chromosomes I, III, IV, and X (*Bazan and Hillers, 2011*; *Wicks et al., 2001*). The SNPs, primers, enzymes used for restriction digests, and expected fragment sizes are included in *Supplementary file 1*. To measure wildtype recombination, N2 males containing *bcIs39* were crossed to Hawaiian CB4856 worms. Cross-progeny hermaphrodites were identified by the presence of *bcIs39* and contained one N2 and one CB4856 chromosome. These were assayed for recombination by crossing with CB4856 males containing *myo-2::mCherry*. Cross-progeny hermaphrodites from the resulting cross were isolated as L4s, and then cultured individually in 96-well plates in liquid S-media complete supplemented with HB101. Four days after initial culturing, starved populations were lysed and used for PCR and restriction digest to detect N2 and CB4856 SNP alleles.

For recombination in *pch-2* mutants, strains homozygous for the CB4856 background of the relevant SNPs were created by backcrossing *pch-2* mutants to worms of the CB4856 background at least eight times and verifying the presence of Hawaiian SNPs on all chromosomes tested in the recombination assay. These Hawaiianized *pch-2* mutants were then mated with *pch-2*; *bcIs39*. Subsequent steps were performed as in the wildtype worms.

### Immunostaining

DAPI staining and immunostaining were performed as in *Russo et al., 2023*, 20–24 hours post L4, unless otherwise noted. For analyzing bivalents, the same protocol was implemented with the exception that hermaphrodites were dissected and DAPI-stained 48 hours post late L4 stage, unless otherwise noted.

The following primary antibodies were used at the indicated dilutions: alpaca anti-GFP Booster (ChromoTek, gb2AF488) was used at 1:1000; rat anti-OLLAS (Invitrogen, PIMA516125) was used at 1:1000; and rabbit anti PCH-2 (*Deshong et al., 2014*) was used at 1:500. The following secondary antibodies were used at the indicated dilutions: anti-rabbit Cy3 (Jackson Labs) was used at 1:500 and anti-rat Cy5 (Jackson Labs) was used at 1:500.

### Irradiation experiments

Control and *pch-2* mutant L4's were aged 12–14 hours before being exposed to 1000 rad (10 Gy) of X-ray radiation using a Precision MultiRad 160 X-irradiator (Precision X-Ray Inc). Germlines were then fixed and stained, 8 hours and 24 hours post irradiation.

### Auxin-induced degradation experiments

Auxin treatment was performed by transferring young adult worms (aged 12–14 hours post-L4) to bacteria-seeded plates containing auxin or 99% ethanol at specific timepoints, except for experiments in which L4s were transferred directly to bacteria-seeded plates containing auxin or ethanol

(the 36 hour and 48 hour timepoints in *Figure 3C and E* and the experiments in *Figure 5C and D*). The natural auxin indole3-acetic acid (IAA) was purchased from Alfa Aesar (#A10556). A 400 mM stock solution in ethanol was prepared and was stored at 4°C for up to 1 month. Auxin was diluted to 100 mM, and 100 ul was spread onto NGM plates. Plates were allowed to dry before seeding with fresh OP50 culture. Plates were left at 20°C for 2–3 days in the dark to allow for bacterial lawn growth.

## Imaging and quantification

All images were acquired using a DeltaVision Personal DV system (Applied Precision) equipped with a ×100 N.A. 1.40 oil-immersion objective (Olympus), resulting in an effective XY pixel spacing of 0.064 or 0.040 μm. Three-dimensional image stacks were collected at 0.2 μm Z spacing and processed by constrained, iterative deconvolution. Image scaling, analysis, and maximum-intensity projections were performed using functions in the softWoRx software package.

For analysis of GFP::MSH-5 foci and meiotic progression, sum projections were generated using ImageJ for each image of the germline. ImageJ plugins Cell Counter, ROI Manager, and Find Maxima were used to identify and quantify GFP::MSH-5 foci by row from transition zone to the end of pachytene. The threshold value was set depending on background conditions to ensure minimal signals were identified. Foci were only quantified if they co-localized with DAPI staining. For all genotypes, three germlines per genotype were analyzed and representative germlines are shown.

## Graphing and statistical analysis

Data was analyzed using Python 3.8 and Prism for statistical significance. All datasets were tested for normality using the Shapiro–Wilk test. For *Figures 1, 2, and 5B*, Fisher's exact test was used to determine significance. For *Figures 3, 4, 5D, and 6*, Mann–Whitney $U$ test was used to determine significance.

## Acknowledgements

We would like to thank Josh Arribere, Pete Carlton, Abby Dernburg, Nicola Silva, and Anne Villeneuve for valuable strains and reagents. We would also like to thank the members of the Bhalla lab for careful review of the manuscript. This work was supported by the NIH (grant numbers R35GM141835 [NB], R25GM051765 [VO] and T34GM140956 [AH and VO]). Some strains were provided by the CGC, which is funded by NIH Office of Research Infrastructure Programs (P40 OD010440).

## Additional information

### Funding

| Funder | Grant reference number | Author |
|---|---|---|
| National Institute of General Medical Sciences | R35GM141835 | Needhi Bhalla |
| National Institute of General Medical Sciences | R25GM051765 | Valery Ortiz |
| National Institute of General Medical Sciences | T34GM140956 | Alberto Herrera Valery Ortiz |

The funders had no role in study design, data collection and interpretation, or the decision to submit the work for publication.

### Author contributions

Bhumil Patel, Conceptualization, Data curation, Formal analysis, Investigation, Visualization, Methodology, Writing – original draft, Writing – review and editing; Maryke Grobler, Conceptualization, Investigation, Methodology, Writing – review and editing; Alberto Herrera, Data curation, Investigation, Writing – review and editing; Elias Logari, Data curation, Investigation; Valery Ortiz, Investigation; Needhi Bhalla, Conceptualization, Resources, Data curation, Formal analysis, Supervision,

Funding acquisition, Validation, Investigation, Visualization, Methodology, Writing – original draft, Project administration, Writing – review and editing

**Author ORCIDs**
Needhi Bhalla (iD) https://orcid.org/0000-0002-6859-0073

Reviewer #1 (Public review): https://doi.org/10.7554/eLife.102409.3.sa1
Reviewer #2 (Public review): https://doi.org/10.7554/eLife.102409.3.sa2
Reviewer #3 (Public review): https://doi.org/10.7554/eLife.102409.3.sa3
Author response https://doi.org/10.7554/eLife.102409.3.sa4

## Additional files

**Supplementary files**
Supplementary file 1. Single-nucleotide polymorphisms (SNPs) used for recombination assay. SNPs, primers, enzymes used for restriction digests, and expected fragment sizes used for the recombination assay.

MDAR checklist

## Data availability
The numerical data used to generate each figure is provided.

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
