## [Editor Report · eLife Assessment]

This is an **important** study examining the role of conserved PCH-2 protein at different stages of *C. elegans* meiosis. The authors use elegant molecular genetic approaches to provide **convincing** evidence to support their claims. The work will be of interest to scientists studying meiosis, DNA recombination, and chromosome segregation.

---

## [Referee Report · Reviewer #1 (Public review)]

The conserved AAA-ATPase PCH-2 has been shown in several organisms including *C. elegans* to remodel classes of HORMAD proteins that act in meiotic pairing and recombination. In some organisms the impact of PCH-2 mutations is subtle but becomes more apparent when other aspects of recombination are perturbed. Patel et al. performed a set of elegant experiments in *C. elegans* aimed at identifying conserved functions of PCH-2. Their work provides such an opportunity because in *C. elegans* meiotically expressed HORMADs localize to meiotic chromosomes independently of PCH-2. Work in *C. elegans* also allows the authors to focus on nuclear PCH-2 functions as opposed to cytoplasmic functions also seen for PCH-2 in other organisms.

The authors performed the following experiments:

(1) They constructed *C. elegans* animals with SNPs that enabled them to measure crossing over in intervals that cover most of four of the six chromosomes. They then showed that double-crossovers, which were common on most of the four chromosomes in wild-type, were absent in pch-2. They also noted shifts in crossover distribution in the four chromosomes.

(2) Based on the crossover analysis and previous studies they hypothesized that PCH-2 plays a role at an early stage in meiotic prophase to regulate how SPO-11 induced double-strand breaks are utilized to form crossovers. They tested their hypothesis by performing ionizing irradiation and depleting SPO-11 at different stages in meiotic prophase in wild-type and pch-2 mutant animals. The authors observed that irradiation of meiotic nuclei in zygotene resulted in pch-2 nuclei having a larger number of nuclei with 6 or greater crossovers (as measured by COSA-1 foci) compared to wildtype. Consistent with this observation, SPO11 depletion, starting roughly in zygotene, also resulted in pch-2 nuclei having an increase in 6 or more COSA-1 foci compared to wildtype. The increased number at this time point appeared beneficial because a significant decrease in univalents was observed.

(3) They then asked if the above phenotypes correlated with the localization of MSH-5, a factor that stabilizes crossover-specific DNA recombination intermediates. They observed that pch-2 mutants displayed an increase in MSH-5 foci at early times in meiotic prophase and an unexpectedly higher number at later times. They conclude based on the differences in early MSH-5 localization and the SPO-11 and irradiation studies that PCH-2 prevents early DSBs from becoming crossovers and early loading of MSH-5. By analyzing different HORMAD proteins that are defective in forming the closed conformation acted upon by PCH-2, they present evidence that MSH-5 loading was regulated by the HIM-3 HORMAD.

(4) They performed a crossover homeostasis experiment in which DSB levels were reduced. The goal of this experiment was to test if PCH-2 acts in crossover assurance. Interestingly, in this background PCH-2 negative nuclei displayed higher levels of COSA-1 foci compared to PCH-2 positive nuclei. This observation and a further test of the model suggested that "PCH-2's presence on the SC prevents crossover designation."

(5) Based on their observations indicating that early DSBS are prevented from becoming crossovers by PCH-2, the authors hypothesized that the DNA damage kinase CHK-2 and PCH-2 act to control how DSBs enter the crossover pathway. This hypothesis was developed based on their finding that PCH-2 prevents early DSBs from becoming crossovers and previous work showing that CHK-2 activity is modulated during meiotic recombination progression. They tested their hypothesis using a mutant synaptonemal complex component that maintains high CHK-2 activity that cannot be turned off to enable crossover designation. Their finding that the pch-2 mutation suppressed the crossover defect (as measured by COSA-1 foci) supports their hypothesis.

Based on these studies the authors provide convincing evidence that PCH-2 prevents early DSBs from becoming crossovers and controls the number and distribution of crossovers to promote a regulated mechanism that ensures the formation of obligate crossovers and crossover homeostasis. As the authors note, such a mechanism is consistent with earlier studies suggesting that early DSBs could serve as "scouts" to facilitate homolog pairing or to coordinate the DNA damage response with repair events that lead to crossing over. The detailed mechanistic insights provided in this work will certainly be used to better understand functions for PCH-2 in meiosis in other organisms.

Comments on revisions:

The authors responded very carefully to all of my concerns expressed in the first review, which were primarily aimed at improving the clarity of the manuscript.

---

## [Referee Report · Reviewer #2 (Public review)]

Summary:

This paper has some intriguing data regarding the different potential roles of Pch-2 in ensuring crossing over. In particular the alterations in crossover distribution and Msh-5 foci are compelling. My main issue is that some of the models are confusingly presented and would benefit from some reframing. The role of Pch-2 across organisms has been difficult to determine, the ability to separate pairing and synapsis roles in worms provides a great advantage for this paper.

Strengths:

Beautiful genetic data, clearly made figures. Great system for studying the role of Pch-2 in crossing over.

Comments on revisions: The authors have responded to all major and minor critiques.

---

## [Referee Report · Reviewer #3 (Public review)]

Summary:

This manuscript describes an in-depth analysis of the effect of the AAA+ ATPase PCH-2 on meiotic crossover formation in C. elegant. The authors reach several conclusions and attempt to synthesize a 'universal' framework for the role of this factor in eukaryotic meiosis.

Strengths:

The manuscript makes use of the advantages of the 'conveyor' belt system within the *C. elegans* reproductive tract, to enable a series of elegant genetic experiments

Weaknesses:

A weakness of this manuscript is that it heavily relies on certain genetic/cell biological assays that can report on distinct crossover outcomes, without clear and directed control over other aspects and variables that might also impact the final repair outcome. Such assays are currently out of reach in this model system.

---

## [Author Response]

The following is the authors’ response to the original reviews.

**Public Reviews:**

**Reviewer #1 (Public review):**
The conserved AAA-ATPase PCH-2 has been shown in several organisms including *C. elegans* to remodel classes of HORMAD proteins that act in meiotic pairing and recombination. In some organisms the impact of PCH-2 mutations is subtle but becomes more apparent when other aspects of recombination are perturbed. Patel et al. performed a set of elegant experiments in *C. elegans* aimed at identifying conserved functions of PCH-2. Their work provides such an opportunity because in *C. elegans* meiotically expressed HORMADs localize to meiotic chromosomes independently of PCH-2. Work in *C. elegans* also allows the authors to focus on nuclear PCH-2 functions as opposed to cytoplasmic functions also seen for PCH-2 in other organisms.The authors performed the following experiments:(1) They constructed *C. elegans* animals with SNPs that enabled them to measure crossing over in intervals that cover most of four of the six chromosomes. They then showed that doublecrossovers, which were common on most of the four chromosomes in wild-type, were absent in pch-2. They also noted shifts in crossover distribution in the four chromosomes.(2) Based on the crossover analysis and previous studies they hypothesized that PCH-2 plays a role at an early stage in meiotic prophase to regulate how SPO-11 induced double-strand breaks are utilized to form crossovers. They tested their hypothesis by performing ionizing irradiation and depleting SPO-11 at different stages in meiotic prophase in wild-type and pch-2 mutant animals. The authors observed that irradiation of meiotic nuclei in zygotene resulted in pch-2 nuclei having a larger number of nuclei with 6 or greater crossovers (as measured by COSA-1 foci) compared to wildtype. Consistent with this observation, SPO11 depletion, starting roughly in zygotene, also resulted in pch-2 nuclei having an increase in 6 or more COSA-1 foci compared to wild type. The increased number at this time point appeared beneficial because a significant decrease in univalents was observed.(3) They then asked if the above phenotypes correlated with the localization of MSH-5, a factor that stabilizes crossover-specific DNA recombination intermediates. They observed that pch-2 mutants displayed an increase in MSH-5 foci at early times in meiotic prophase and an unexpectedly higher number at later times. They conclude based on the differences in early MSH-5 localization and the SPO-11 and irradiation studies that PCH-2 prevents early DSBs from becoming crossovers and early loading of MSH-5. By analyzing different HORMAD proteins that are defective in forming the closed conformation acted upon by PCH-2, they present evidence that MSH-5 loading was regulated by the HIM-3 HORMAD.(4) They performed a crossover homeostasis experiment in which DSB levels were reduced. The goal of this experiment was to test if PCH-2 acts in crossover assurance. Interestingly, in this background PCH-2 negative nuclei displayed higher levels of COSA-1 foci compared to PCH-2 positive nuclei. This observation and a further test of the model suggested that "PCH-2's presence on the SC prevents crossover designation."(5) Based on their observations indicating that early DSBS are prevented from becoming crossovers by PCH-2, the authors hypothesized that the DNA damage kinase CHK-2 and PCH2 act to control how DSBs enter the crossover pathway. This hypothesis was developed based on their finding that PCH-2 prevents early DSBs from becoming crossovers and previous work showing that CHK-2 activity is modulated during meiotic recombination progression. They tested their hypothesis using a mutant synaptonemal complex component that maintains high CHK-2 activity that cannot be turned off to enable crossover designation. Their finding that the pch-2 mutation suppressed the crossover defect (as measured by COSA-1 foci) supports their hypothesis.Based on these studies the authors provide convincing evidence that PCH-2 prevents early DSBs from becoming crossovers and controls the number and distribution of crossovers to promote a regulated mechanism that ensures the formation of obligate crossovers and crossover homeostasis. As the authors note, such a mechanism is consistent with earlier studies suggesting that early DSBs could serve as "scouts" to facilitate homolog pairing or to coordinate the DNA damage response with repair events that lead to crossing over. The detailed mechanistic insights provided in this work will certainly be used to better understand functions for PCH-2 in meiosis in other organisms. My comments below are aimed at improving the clarity of the manuscript.

We thank the reviewer for their concise summary of our manuscript and their assessment of our work as “convincing” and providing “detailed mechanistic insight.”

Comments(1) It appears from reading the Materials and Methods that the SNPs used to measure crossing over were obtained by mating Hawaiian and Bristol strains. It is not clear to this reviewer how the SNPs were introduced into the animals. Was crossing over measured in a single animal line? Were the wild-type and pch-2 mutations made in backgrounds that were isogenic with respect to each other? This is a concern because it is not clear, at least to this reviewer, how much of an impact crossing different ecotypes will have on the frequency and distribution of recombination events (and possibly the recombination intermediates that were studied).

We have clarified these issues in the Materials and Methods of our updated preprint. The control and *pch-2* mutants were isogenic in either the Bristol or Hawaiian backgrounds. Control lines were the original Bristol and Hawaiian lines and *pch-2* mutants were originally made in the Bristol line and backcrossed at least 3 times before analysis. Hawaiian *pch-2* mutants were made by backcrossing *pch-2* mutants at least 8 times to the Hawaiian background and verifying the presence of Hawaiian SNPs on all chromosomes tested in the recombination assay. To perform the recombination assays, these lines were crossed to generate the relevant F1s.

(2) The authors state that in pch-2 mutants there was a striking shift of crossovers (line 135) to the PC end for all of the four chromosomes that were tested. I looked at Figure 1 for some time and felt that the results were more ambiguous. Map distances seemed similar at the PC end for wildtype and pch-2 on Chrom. I. While the decrease in crossing over in pch-2 appeared significant for Chrom. I and III, the results for Chrom. IV, and Chrom. X. seemed less clear. Were map distances compared statistically? At least for this reviewer the effects on specific intervals appear less clear and without a bit more detail on how the animals were constructed it's hard for me to follow these conclusions.

We hope that the added details above makes the results of these assays more clear. Map distances were compared and did not satisfy statistical significance, except where indicated. While we agree that the comparisons between control animals and *pch-2* mutants may seem less clear with individual chromosomes, we argue that more general, consistent patterns become clear when analyzing multiple chromosomes. Indeed, this is why we expanded our recombination analysis beyond Chromosome III and the X Chromosome, as reported in Deshong, 2014. We have edited this sentence to: “Moreover, there was a striking and consistent shift of crossovers to the PC end of all four chromosomes tested.”

(3) Figure 2. I'm curious why non-irradiated controls were not tested side-by-side for COSA-1 staining. It just seems like a nice control that would strengthen the authors' arguments.

We have added these controls in the updated preprint as Figure 2B.

(4) Figure 3. It took me a while to follow the connection between the COSA-1 staining and DAPI staining panels (12 hrs later). Perhaps an arrow that connects each set of time points between the panels or just a single title on the X-axis that links the two would make things clearer.

To make this figure more clear, we have generated two different cartoons for the assay that scores GFP::COSA-1 foci and the assay that scores bivalents. We have also edited this section of the results to make it more clear.

**Reviewer #2 (Public review):**
Summary:This paper has some intriguing data regarding the different potential roles of Pch-2 in ensuring crossing over. In particular, the alterations in crossover distribution and Msh-5 foci are compelling. My main issue is that some of the models are confusingly presented and would benefit from some reframing. The role of Pch-2 across organisms has been difficult to determine, the ability to separate pairing and synapsis roles in worms provides a great advantage for this paper.Strengths:Beautiful genetic data, clearly made figures. Great system for studying the role of Pch-2 in crossing over.

We thank the reviewers for their constructive and useful summary of our manuscript and the analysis of its strengths.

Weaknesses:(1) For a general audience, definitions of crossover assurance, crossover eligible intermediates, and crossover designation would be helpful. This applies to both the proposed molecular model and the cytological manifestation that is being scored specifically in *C. elegans*.

We have made these changes in an updated preprint.

(2) Line 62: Is there evidence that DSBs are introduced gradually throughout the early prophase? Please provide references.

We have referenced Woglar and Villeneuve 2018 and Joshi et. al. 2015 to support this statement in the updated preprint.

(3) Do double crossovers show strong interference in worms? Given that the PC is at the ends of chromosomes don't you expect double crossovers to be near the chromosome ends and thus the PC?

Despite their rarity, double crossovers do show interference in worms. However, the PC is limited to one end of the chromosome. Therefore, even if interference ensures the spacing of these double crossovers, the preponderance of one of these crossovers toward one end (and not both ends) suggest something functionally unique about the PC end.

(4) Line 155 - if the previous data in Deshong et al is helpful it would be useful to briefly describe it and how the experimental caveats led to misinterpretation (or state that further investigation suggests a different model etc.). Many readers are unlikely to look up the paper to find out what this means.

We have added this to the updated preprint: “We had previously observed that meiotic nuclei in early prophase were more likely to produce crossovers when DSBs were induced by the *Mos* transposon in *pch-2* mutants than in control animals but experimental caveats limited our ability to properly interpret this experiment.”

(5) Line 248: I am confused by the meaning of crossover assurance here - you see no difference in the average number of COSA-1 foci in Pch-2 vs. wt at any time point. Is it the increase in cells with >6 COSA-1 foci that shows a loss of crossover assurance? That is the only thing that shows a significant difference (at the one time point) in COSA-1 foci. The number of dapi bodies shows the loss of Pch-2 increases crossover assurance (fewer cells with unattached homologs). So this part is confusing to me. How does reliably detecting foci vs. DAPI bodies explain this?

We have removed this section to avoid confusion.

(6) Line 384: I am confused. I understand that in the dsb-2/pch2 mutant there are fewer COSA-1 foci. So fewer crossovers are designated when DSBs are reduced in the absence of PCH-2.How then does this suggest that PCH-2's presence on the SC prevents crossover designation? Its absence is preventing crossover designation at least in the dsb-2 mutant.

We have tried to make this more clear in the updated preprint. In this experiment, we had identified three possible explanations for why PCH-2 persists on some nuclei that do not have GFP::COSA-1 foci: (1) PCH-2 removal is coincident with crossover designation; (2) PCH-2 removal depends on crossover designation; and (3) PCH-2 removal facilitates crossover designation. The decrease in the number of GFP::COSA-1 foci in *dsb2::AID;pch-2* mutants argues against the first two possibilities, suggesting that the third might be correct. We have edited the sentence to read: “These data argue against the possibility that PCH-2’s removal from the SC is simply in response to or coincident with crossover designation and instead, suggest that PCH-2’s removal from the SC somehow facilitates crossover designation and assurance.”

(7) Discussion Line 535: How do you know that the crossovers that form near the PCs are Class II and not the other way around? Perhaps early forming Class I crossovers give time for a second Class II crossover to form. In budding yeast, it is thought that synapsis initiation sites are likely sites of crossover designation and class I crossing over. Also, the precursors that form class I and II crossovers may be the same or highly similar to each other, such that Pch-2's actions could equally affect both pathways.

We do not know that the crossovers that form near the PC are Class II but hypothesize that they are based on the close, functional relationship that exists between Class I crossovers and synapsis and the apparent antagonistic relationship that exists between Class II crossovers and synapsis. We agree that Class I and Class II crossover precursors are likely to be the same or highly similar, exhibit extensive crosstalk that may complicate straightforward analysis and PCH-2 is likely to affect both, as strongly suggested by our GFP::MSH-5 analysis. We present this hypothesis based on the apparent relationship between PCH-2 and synapsis in several systems but agree that it needs to be formally tested. We have tried to make this argument more clear in the updated preprint.

**Reviewer #3 (Public review):**
Summary:This manuscript describes an in-depth analysis of the effect of the AAA+ ATPase PCH-2 on meiotic crossover formation in C. elegant. The authors reach several conclusions, and attempt to synthesize a 'universal' framework for the role of this factor in eukaryotic meiosis.Strengths:The manuscript makes use of the advantages of the 'conveyor' belt system within the *C. elegans* reproductive tract, to enable a series of elegant genetic experiments.

We thank this reviewer for the useful assessment of our manuscript and the articulation of its strengths.

Weaknesses:A weakness of this manuscript is that it heavily relies on certain genetic/cell biological assays that can report on distinct crossover outcomes, without clear and directed control over other aspects and variables that might also impact the final repair outcome. Such assays are currently out of reach in this model system.In general, this manuscript could be more generally accessible to non-*C. elegans* readers. Currently, the manuscript is hard to digest for non-experts (even if meiosis researchers). In addition, the authors should be careful to consider alternative explanations for certain results. At several steps in the manuscript, results could ostensibly be caused by underlying defects that are currently unknown (for example, can we know for sure that pch-2 mutants do not suffer from altered DSB patterning, and how can we know what the exact functional and genetic interactions between pch-2 and HORMAD mutants tell us?). Alternative explanations are possible and it would serve the reader well to explicitly name and explain these options throughout the manuscript.

We have made the manuscript more accessible to non-*C. elegans* readers and discuss alternate explanations for specific results in the updated preprint.

**Recommendations for the authors:**

**Reviewing Editor Comments:**
(1) Please provide 'n' values for each experiment.

n values are now included in the Figure legends for each experiment.

(2) Line 129: Please represent the DCOs as percent or fraction (1%-9.8%, instead of 1-13).

We have made this change.

(3) Figure 3A legend: the grey bar should read 20hr. COSA-1/ 32 hr DAPI. In Figure 3E, it is not clear why 36hr Auxin and 34hr Auxin show a significant difference in DAPI bodies between control and pch-2, but 32hr Auxin treatment does not. Here again 'n' values will help.

We have made this change. We also are not sure why the 32 hour auxin treatment did not show a significant difference in DAPI stained bodies. We have included the n values, which are not very different between timepoints and therefore are unlikely to explain the difference. The difference may reflect the time that it takes for SPO-11 function to be completely abrogated.

(4) Line 360: Please provide the fraction of PCH-2 positive nuclei in dsb-2.

We have made this change.

Please also address all reviewer comments.
**Reviewer #1 (Recommendations for the authors):**
(1) Page 3, line 52. While I agree that crossing over is important to generate new haplotypes, work has suggested that the contribution by an independent assortment of homologs to generate new haplotypes is likely to be significantly greater. One reference for this is: Veller et al. PNAS 116:1659.

We deeply appreciate this reviewer pointing us to this paper, especially since it argues that controlling crossover distribution contributes to gene shuffling and now cite it in our introduction! While we agree that this paper concludes that independent assortment likely explains the generation of new haplotypes to a greater degree than crossovers, the authors performed this analysis with human chromosomes and explicitly include the caveat that their modeling assumes uniform gene density across chromosomes. For example, we know this is not true in *C. elegans*. It would be interesting to perform the same analysis with *C. elegans* chromosomes in control and *pch-2* mutants, taking into account this important difference.

(2) Figure 2. It would really help the reader if an arrow and text were shown below each irradiation sign to indicate the stage in meiosis in which the irradiation was done as well as another arrow in the late pachytene box to show when the COSA-1 foci were analyzed. In general, having text in the figures that help stage the timing in meiosis would help the non *C. elegans* reader. This is also an issue where staging of *C. elegans* is shown (Figure 4).

We have made these changes to Figure 2. To help readers interpret Figure 4, we have added TZ and LP to the graphs in Figure 4B and 4D and indicated what these acronyms (transition zone and late pachytene, respectively) are in the Figure legend.

(3) Page 12, line 288. It would be valuable to first outline why the him3-R93Y and htp-3H96Y alleles were chosen. This was eventually done on Page 13, but introducing this earlier would help the reader.

We have introduced these mutations earlier in the manuscript.

(4) Page 13, line 323. A one sentence description of the OLLAS tagging system would be useful.

We have added this sentence: “we generated wildtype animals and *pch-2* mutants with both GFP::MSH-5 and a version of COSA-1 that has been endogenously tagged at the Nterminus with the epitope tag, OLLAS, a fusion of the *E. coli* OmpF protein and the mouse Langerin extracellular domain”

**Reviewer #2 (Recommendations for the authors):**
(1) The title is a little awkward. Consider: PCH-2 controls the number and distribution of crossovers in *C. elegans* by antagonizing their formation

We have made this change.

(2) Abstract:Consider removing "that is observed" from line 20.

We have made this change.

I'm confused by the meaning of "reinforcement of crossover-eligible intermediates" from line 27.

We have removed this phrase from the abstract.

A definition of crossover assurance would be helpful in the abstract.

We have added this to the abstract: “This requirement is known as crossover assurance and is one example of crossover control.”

(3) Line 36: I know a stickler but many meioses only produce one haploid gamete (mammalian oocytes, for example)

Thanks for the reminder! We have removed the “four” from this sentence.

(4) Line 284 - are you defining MSH-5 foci as crossover-eligible intermediates? If so, please state this earlier.

We have added this to the introduction to this section of the results: “In *C. elegans*, these crossover-eligible intermediates can be visualized by the loading of the pro-crossover factor MSH-5, a component of the meiosis-specific MutSγ complex that stabilizes crossover-specific DNA repair intermediates called joint molecules”

(5) Can the control be included in Figure S1?

We have made this change.

(6) Can you define that crossover designation is the formation of a COSA-1 focus?

We did this in the section introducing GFP::MSH-5: “In the spatiotemporally organized meiotic nuclei of the germline, a functional GFP tagged version of MSH-5, GFP::MSH-5, begins to form a few foci in leptotene/zygotene (the transition zone), becoming more numerous in early pachytene before decreasing in number in mid pachytene to ultimately colocalize with COSA-1 marked sites in late pachytene in a process called designation”

(7) Would it be easier to see the effect of DSB to crossover eligible intermediates in Spo-11, Pch-2 vs. Spo-11 mutant with irradiation using your genetic maps? At least for early vs. late breaks?

Unfortunately, irradiation does not show the same bias towards genomic location that endogenous double strand breaks do so it is unlikely to recapitulate the effects on the genetic map.